# PI16$^+$ reticular cells in human palatine tonsils govern T cell activity in distinct subepithelial niches

Angelina De Martin[1,4], Yves Stanossek[1,2,4], Mechthild Lütge [1], Nadine Cadosch[1], Lucas Onder[1], Hung-Wei Cheng[1], Joshua D. Brandstadter [3], Ivan Maillard [3], Sandro J. Stoeckli[2], Natalia B. Pikor [1] & Burkhard Ludewig [1] ✉

Fibroblastic reticular cells (FRCs) direct the interaction and activation of immune cells in discrete microenvironments of lymphoid organs. Despite their important role in steering innate and adaptive immunity, the age- and inflammation-associated changes in the molecular identity and functional properties of human FRCs have remained largely unknown. Here, we show that human tonsillar FRCs undergo dynamic reprogramming during life and respond vigorously to inflammatory perturbation in comparison to other stromal cell types. The peptidase inhibitor 16 (PI16)-expressing reticular cell (PI16$^+$ RC) subset of adult tonsils exhibited the strongest inflammation-associated structural remodeling. Interactome analysis combined with ex vivo and in vitro validation revealed that T cell activity within subepithelial niches is controlled by distinct molecular pathways during PI16$^+$ RC–lymphocyte interaction. In sum, the topological and molecular definition of the human tonsillar stromal cell landscape reveals PI16$^+$ RCs as a specialized FRC niche at the core of mucosal immune responses in the oropharynx.

Activation and differentiation of T and B cell responses in the immune system are controlled in lymphoid organs and lymphoid tissue structures, which increase the odds for antigen encounter by rare naive T or B cell clones and optimize their interaction[1–3]. The major secondary lymphoid organs (SLOs) underpinning the mammalian immune system comprise the splenic white pulp, lymph nodes and Peyer's patches that are generated through conserved molecular and cellular interactions[4,5]. Partially overlapping, yet distinct developmental pathways have secured the establishment of specialized subsystems within SLOs to sample antigen and to facilitate immunological surveillance of bodily (for example, lymph or blood) or external fluids such as the content of the intestinal tract[3,6–9]. The human oropharynx is lined by a series of different SLOs known as adenoid, tubal, lingual and palatine tonsils, which

all together form the ring of Waldeyer[10,11]. The position of the palatine tonsils at the gateway of both the respiratory tract and the digestive tract leads to constant exposure to microorganisms including a wide range of commensal bacteria[12,13] or pathogenic viruses[14,15]. Antigenic material including pathogens and commensal bacteria gains access to the lymphoid tissue after passage through a specialized epithelium that covers deep invaginations (crypts) of the tonsil[10,11]. However, it is still unclear how the specialized crypt epithelium relates to and communicates with the underlying lymphoid tissues.

Communication between immune cells within particular SLO compartments is organized by lymphoid organ fibroblasts that are commonly referred to as FRCs[9,16]. Specialized FRC subsets form the scaffold and determine the microenvironmental conditions for lymphocyte

[1]Institute of Immunobiology, Kantonsspital St. Gallen, St. Gallen, Switzerland. [2]Department of Otorhinolaryngology, Head and Neck Surgery, Kantonsspital St. Gallen, St. Gallen, Switzerland. [3]Division of Hematology–Oncology, Department of Medicine, University of Pennsylvania Perelman School of Medicine, Philadelphia, PA, USA. [4]These authors contributed equally: Angelina De Martin, Yves Stanossek. ✉e-mail: burkhard.ludewig@kssg.ch

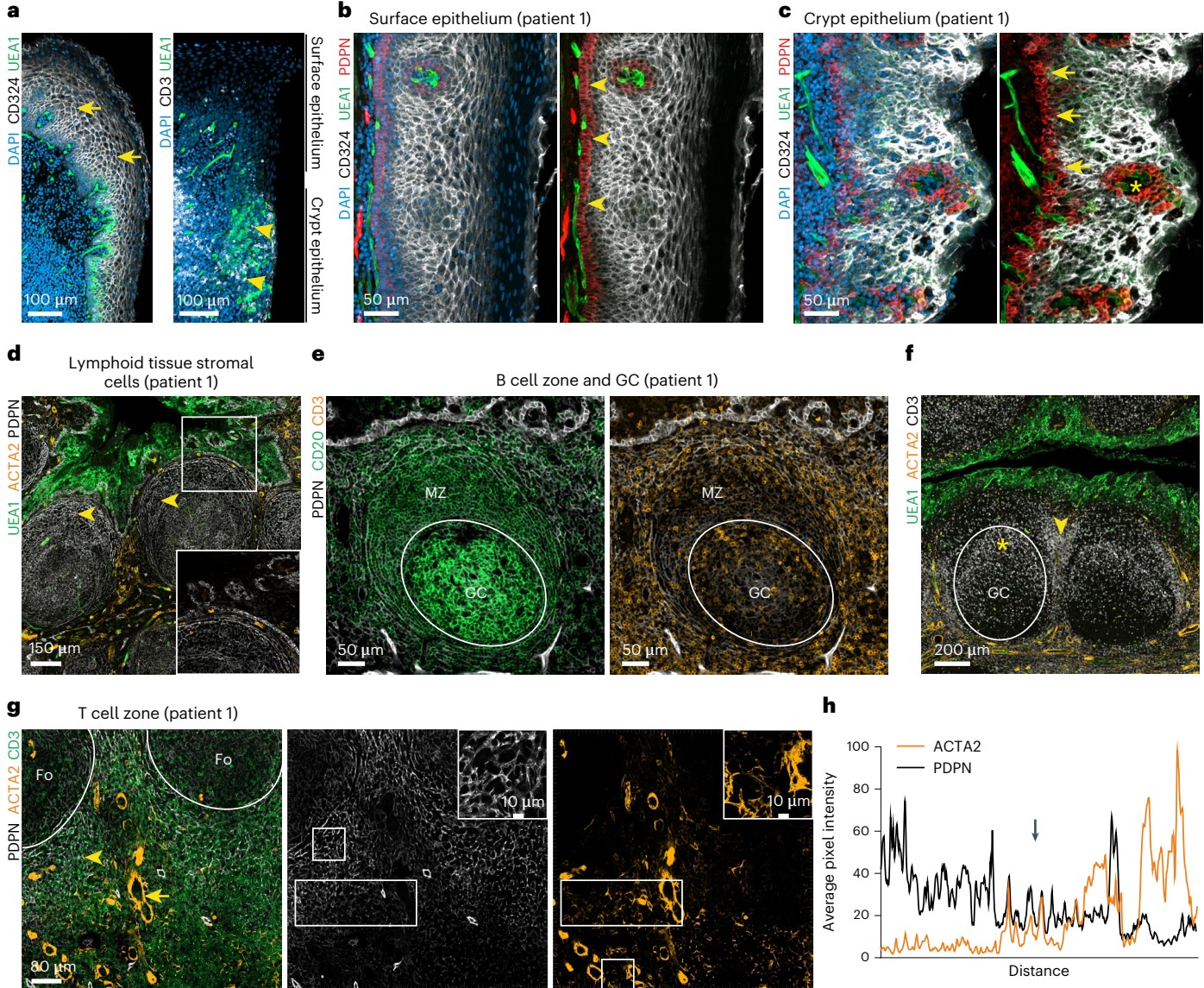

**Fig. 1 | Stromal cell topology in human palatine tonsils. a–g**, Representative immunofluorescence images of palatine tonsil sections from adult patients with OSA stained with the indicated antibodies and analyzed by confocal microscopy. **a**, Transition zone from the surface to the crypt epithelium. Stratified squamous epithelial cells at the surface (arrows) and UEA1[+] epithelial cells infiltrated with CD3[+] lymphocytes (arrowheads) at the transition area to the crypt epithelium are highlighted. DAPI, 4,6-diamidino-2-phenylindole. **b**, High-resolution analysis of the non-keratinized stratified squamous surface epithelium. PDPN-expressing basal cuboidal epithelial cells are highlighted (arrowheads). **c**, Morphology of the lymphoreticular crypt epithelium. Broadened PDPN-expressing basal layer of the crypt epithelium (arrows) and papillary extension providing access for blood vessels (asterisk) are highlighted. **d**, Epithelial and subepithelial (arrowheads) areas underpinned by PDPN[+] stromal cells. Boxed area shows the magnified epithelial–lymphoid tissue interface. **e**, B cell follicles with GCs and mantle zones (MZ) underpinned by PDPN[+] FRCs. **f**, CD3[+] lymphocytes in epithelial, GC (asterisk) and interfollicular T cell (arrowhead) areas. **g**, T cell area surrounding B cell follicles (Fo) underpinned by PDPN[+] FRCs (arrowhead) and ACTA2[+]PDPN[−] VSMCs (arrow). Square boxes show cell networks at higher magnification, and rectangular boxes indicate the regions of PDPN- and ACTA2-intensity measurements shown in **h**. **h**, Average pixel intensity of ACTA2 and PDPN signal-intensity measurements. The x axis represents the horizontal distance through the selection (rectangular boxes in **g**), and the y axis represents the vertically averaged pixel intensity. The arrow highlights ACTA2[+]PDPN[+] myofibroblasts in the perivascular space. Microscopy images are representative for n = 3 adult patients with OSA.

guidance, activation and differentiation[17,18]. Over the course of pathogen encounter and immune activation, the FRC topology profoundly changes to accommodate increased numbers of immune cells and to support efficient immune cell interaction[19,20]. For example, murine FRCs govern intestinal microbiome homeostasis and support anti-coronavirus B cell responses[7], regulate anti-coronavirus group 1 innate lymphoid cell function[21], support the inflation of memory CD8[+] T cells after adenovirus vector vaccination[22] and control innate

lymphoid cell homeostasis and activation in small intestinal lymphoid tissues[23]. Here, we have combined single-cell RNA-sequencing (scRNA-seq) analysis with flow cytometry and high-resolution confocal microscopy to resolve the human tonsillar FRC landscape and to determine major interaction patterns with immune cells. We found that perivascular and subepithelial FRC subsets undergo the most dynamic age- and inflammation-associated transcriptional reprogramming and topological remodeling. Subepithelial PI16[+] RCs formed a distinct niche

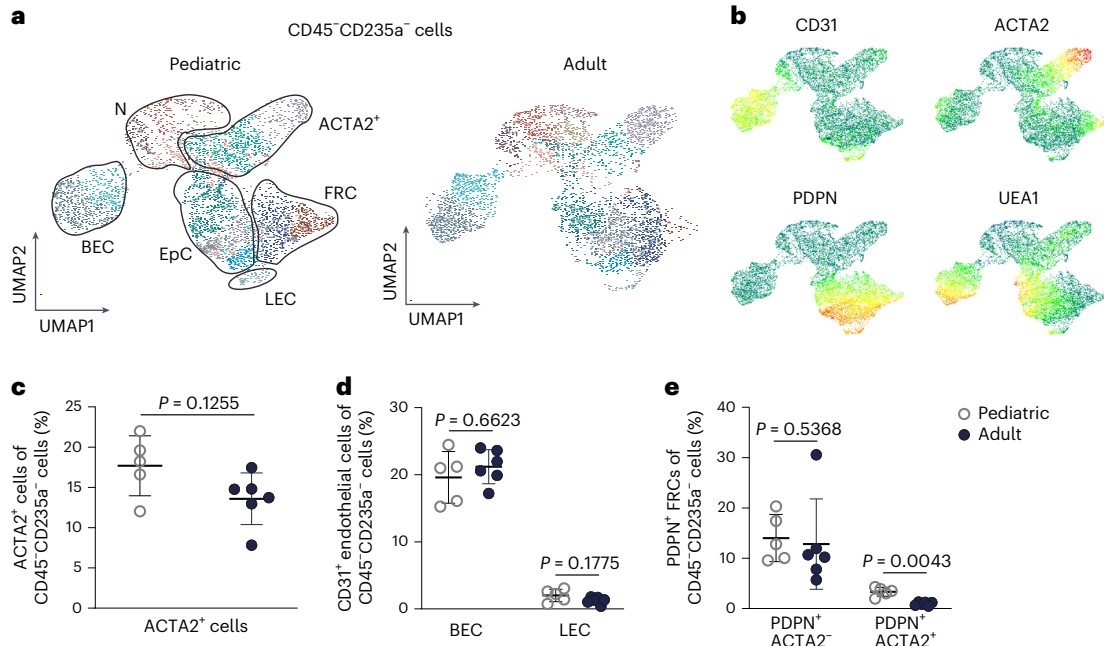

**Fig. 2 | Age-associated changes of the tonsillar stromal cell landscape.**
**a**, Representative UMAPs of distinct tonsillar stromal cell types according to the PhenoGraph clustering algorithm based on forward scatter (FSC)-A, side scatter (SSC)-A, CD31, ACTA2, PDPN and UEA1 flow cytometry data of CD45⁻CD235a⁻ cells (*n* = 3 pediatric and *n* = 3 adult patients with OSA). Epithelial cells (EpCs), ACTA2⁺ cells, BECs, LECs, FRCs and negative cells (N) are indicated. **b**, UMAPs show the expression pattern of the indicated markers. **c**–**e**, Quantification of the indicated stromal cell types as a percentage of CD45⁻CD235a⁻ live cells in pediatric (*n* = 5) and adult (*n* = 6) patients with OSA. Data show average values of left and right tonsils for each patient. Mean and s.d. are indicated. *P* values were calculated with the two-sided Mann–Whitney test.

adjacent to the lymphoreticular epithelium in adult tonsils supporting direct contact with T and B cells and the provision of co-stimulatory and growth factors that regulate T cell activity. Overall, the study indicates a key position of FRCs in the basic organization, adaptive remodeling and molecular control of mucosal immune processes in the human oropharynx.

## Results

### Topology of the tonsillar stromal cell landscape

Palatine tonsils are characterized by deep ramified crypts that increase the surface area for antigen capture and transfer through the epithelial layer to the underlying lymphoid tissue (Extended Data Fig. 1a). To demarcate different stromal cell populations, we used antibodies to α-smooth muscle actin (ACTA)2 to highlight myofibroblasts, CD324 (E-cadherin) to stain for epithelial cells and the lectin *Ulex europaeus* agglutinin 1 (UEA1) for the detection of epithelial and blood endothelial cells (BECs; Fig. 1a and Extended Data Fig. 1a). Infiltration of the crypt epithelium with immune cells (Fig. 1a, arrowheads) was associated with UEA1 binding by epithelial cells, whereas the densely packed surface epithelium was negative for UEA1 (Fig. 1a, arrows). As a large fraction of lymphoid organ fibroblast subsets express the surface glycoprotein podoplanin (PDPN)[16,17], we used this marker to highlight tonsillar FRCs. We found that the basal layer of CD324⁺ surface epithelial cells exhibited high PDPN expression (Fig. 1b and Extended Data Fig. 1b,d, arrowheads). Likewise, the broadened basal layer of the crypt epithelium showed strong PDPN expression (Fig. 1c and Extended Data Fig. 1c,e, arrows). The lymphoreticular crypt epithelium was pervaded by papillary extensions of the basal layer providing access for UEA1⁺ blood vessels (Fig. 1c and Extended Data Fig. 1e, asterisk) to the immune cell-rich compartment that was occupied by CD11c⁺ myeloid cells (Extended Data Fig. 1f). The PDPN⁺ basal layer of cuboidal epithelial cells was clearly distinguishable from the follicular PDPN⁺ fibroblast network (Fig. 1d and Extended Data Fig. 2a,d, arrowheads). The follicular structures

were mainly occupied by B cells in germinal centers (GCs) containing CD20ʰⁱ B cells and an apical mantle zone (Fig. 1e and Extended Data Fig. 2b,e, left). CD3⁺ T cells were more scarce in the B cell follicle, except for the apical zone of the GC (Fig. 1e and Extended Data Fig. 2b,e, right; Fig. 1f and Extended Data Fig. 2c,f, asterisk). T cells were abundant in the interfollicular regions (Fig. 1f and Extended Data Fig. 2c,f, arrowhead) and the blood vessel-rich central zone (Fig. 1g and Extended Data Fig. 2g,h). The central T cell zone was underpinned by three fibroblast populations: ACTA2⁺PDPN⁻ vascular smooth muscle cells (VSMCs; Fig. 1g and Extended Data Fig. 2g,h, arrow), PDPN⁺ACTA2⁻ fibroblasts forming the large part of the T cell zone fibroblast network (Fig. 1g and Extended Data Fig. 2g,h, arrowhead) and a smaller population of ACTA2⁺PDPN⁺ fibroblasts in the perivascular space (Fig. 1g and Extended Data Fig. 2g,h). Pixel-intensity analysis confirmed the inverse gradients of PDPN and ACTA2 staining around larger blood vessels yielding a distinct perivascular FRC compartment (Fig. 1h, arrow). In sum, these data highlight the regional organization and heterogeneity of the human tonsillar stromal cell landscape.

### Age-associated changes in tonsillar stromal cell populations

To assess to what extent age affects the tonsillar stromal cell landscape, we determined the stromal cell composition in two cohorts of children and adults diagnosed with obstructive sleep apnea (OSA) without clinical signs of inflammation. Based on our microscopy analysis, we established a flow cytometric antibody or marker panel to distinguish the non-hematopoietic (CD45⁻), non-erythrocytic (CD235a⁻) cells including CD31⁻UEA1⁺ lymphoreticular epithelial cells, ACTA2⁺ myofibroblastic cells, CD31⁺UEA1⁺PDPN⁻ BECs, CD31⁺PDPN⁺ lymphatic endothelial cells (LECs) and PDPN⁺UEA1⁻ FRCs (Extended Data Fig. 3a,b). Uniform manifold approximation and projection (UMAP) visualization revealed three populations of epithelial cells that differed in PDPN expression (Fig. 2a,b). The fractions of ACTA⁺ cells (Fig. 2c) and epithelial cells (Extended Data Fig. 3c) among non-hematopoietic cells

did not differ between pediatric and adult patients with OSA. Likewise, endothelial cell composition in patients with OSA was not affected with comparable fractions of LECs and BECs in pediatric patients compared to adult patients (Fig. 2d). The large fraction of PDPN⁺ACTA2⁻ FRCs was not affected by the age differences in patients with OSA, whereas PDPN⁺ACTA2⁺ perivascular fibroblasts were significantly more abundant in pediatric tonsils (Fig. 2e). The panel used in this analysis did not permit assignment of all non-hematopoietic cells to clearly defined cell populations, leaving 17.9% ± 4% of the cells in a marker-negative fraction. Nevertheless, the four-marker-based approach permits the resolution of the major stromal cell types in human palatine tonsils and revealed age-dependent changes in the tonsillar FRC landscape.

## Molecular definition of tonsillar stromal cell populations

To distinguish age- and inflammation-dependent molecular changes in tonsillar stromal cells, we sorted CD45⁻CD235a⁻ cells from pediatric and adult OSA tonsils (Fig. 3a and Extended Data Fig. 3d) and from acutely or chronically inflamed tonsils (Extended Data Table 1). Using droplet-based single-cell transcriptomics, 86,966 tonsillar stromal cells were profiled (20,158 cells from pediatric patients with OSA, 21,596 cells from adult patients with OSA and 45,212 cells from adult patients with tonsillitis). Unsupervised clustering and UMAP visualization yielded 14 stromal cell clusters (Fig. 3b). Computation of cluster-specific genes revealed transcriptional signatures consistent with BECs, LECs, FRCs, smooth muscle cells, *ACTA2*-expressing cells, skeletal muscle cells and epithelial cells (Fig. 3b,c and Extended Data Fig. 3e–g). A substantial fraction of epithelial cells in clusters 12 and 13 were positive for *PDPN* (Fig. 3c) and thus represent epithelial cells in the basal layer (Fig. 1b,c and Extended Data Fig. 1b–e). *ACTA2*⁺ cells in cluster 9 could be further classified as VSMCs based on the high expression of *MCAM*, *MYH11*, *TPM1* and *TAGLN*[24,25]. The second *ACTA2*⁺ cell population (cluster 8) showed lower *ACTA2* expression and revealed a substantial overlap with FRC signature genes, including *THY1*, *PDGFRB*, *COL1A2*, and *CCL19* and *CCL21*, encoding chemokines (Fig. 3c). Projection of VSMC (Fig. 3d) and FRC gene signatures (Fig. 3e) onto the transcriptional tonsillar stromal cell landscape and computation of shared differentially expressed genes (Fig. 3f) revealed *ACTA2*⁺ cells in cluster 8 as perivascular reticular cells (PRCs). Gene set enrichment analysis based on shared differentially expressed genes indicated that *ACTA2*⁺ PRCs exhibit classical FRC functions (Fig. 3g,h). Gene set enrichment analysis based on genes shared between *ACTA2*⁺ PRCs and VSMCs supported the notion that reticular cells in the perivascular space actively shape the topology of the lymphoid tissue and that the cells are fueled by efficient metabolic processes including oxidative phosphorylation (Fig. 3i,j). In sum, the comprehensive molecular characterization and definition of tonsillar stromal cells reveals *ACTA2*⁺ PRCs as one of the main FRC subsets that contributes to adaptive remodeling of lymphoid tissues.

## PI16⁺ RCs interact with lymphocytes in a subepithelial niche

Next, we compared the transcriptomes of CD45⁻CD235a⁻ cells from pediatric patients with OSA and adult patients with OSA or tonsillitis (Fig. 4a–c). The transcriptional profile within FRC populations changed profoundly in adult tonsils (Fig. 4b,c, arrows) compared to pediatric tonsils (Fig. 4a). Moreover, we found that the relative population size of FRCs increased with age and inflammatory perturbation (Fig. 4d and Extended Data Fig. 4a), indicating that both age and inflammation lead to substantial remodeling of the tonsillar FRC compartment.

Reanalysis of the age- and inflammation-associated changes in the FRC transcriptome revealed 12 different clusters at the pre-defined resolution (Fig. 4e). Evaluation of cluster-specific gene expression facilitated grouping into four major FRC subsets: ACTA2⁺ PRCs, follicular dendritic cells, T cell zone reticular cells (TRCs) and a population highlighted by the expression of *PI16*, a gene that encodes peptidase

inhibitor 16 (PI16) (ref. 26) (Extended Data Fig. 4e). Markers highly enriched in ACTA2⁺ PRCs included *RGS5* (encoding the pericyte marker NG-2), *MCAM* (encoding the surface glycoprotein CD146) and *CCL19*, encoding a chemokine (Fig. 4f and Extended Data Fig. 4f). Follicular dendritic cells expressed a set of specific genes including the complement receptor genes *CR1* and *CR2* (Fig. 4f and Extended Data Fig. 4e). By contrast, gene expression in the TRC populations appeared to be more heterogeneous (Fig. 4f and Extended Data Fig. 4e,f), most likely reflecting the chemokine and growth factor gradients between the central T cell zone and the T–B cell zone border[27,28]. A large fraction of FRCs shared the expression of *PI16*, *CD34* (encoding a transmembrane sialomucin expressed on hematopoietic stem cells) and the gene encoding dipeptidyl peptidase 4 (*DPP4*) (Fig. 4f and Extended Data Fig. 4e), which is coexpressed with *PI16* in a particular fibroblast subset present in a multitude of tissues[29]. Here, we found that the presence of *PI16*⁺ RCs was almost completely restricted to adult tonsils and that inflammation substantially affected the transcriptional phenotype of this particular FRC subset (Fig. 4g and Extended Data Fig. 4g). To further substantiate the almost complete absence of *PI16*⁺ RCs in tonsils of pediatric patients, we incorporated three independently acquired and processed scRNA-seq datasets (Extended Data Table 1). Integration of both datasets using batch correction approaches (Extended Data Fig. 4b,c) confirmed the low abundance of *PI16*⁺ RCs in tonsils of pediatric patients (Extended Data Fig. 4d). Gene set enrichment analysis based on all differentially expressed genes among the four FRC subsets indicated that *PI16*⁺ RCs exhibited an array of different functions (Extended Data Fig. 4h). Projection of the functional gene sets on the diffusion map, which permits inference on developmental and differentiation processes by ordering the cells according to their transcriptional state, highlighted the fact that the inflammatory conditions present in patients with tonsillitis strongly affect *PI16*⁺ RC functions (Fig. 4h,i).

Confocal microscopy analysis on cryopreserved sections confirmed that the intracellular peptidase PI16 was expressed by only a few cells in pediatric patients' tonsils (Extended Data Fig. 5c,d), whereas PDPN⁺ FRCs in adult patients with OSA (Fig. 5a and Extended Data Fig. 5a) and adult patients with tonsillitis (Fig. 5b and Extended Data Fig. 5b) showed a higher abundance of PI16-expressing cells. Three-dimensional (3D) reconstruction of stacked images showed perinuclear localization of the protein (Fig. 5c and Extended Data Fig. 5e–g). Consistent with gene expression data (Fig. 4e), PI16⁺ RCs coexpressed the secreted glycoprotein fibulin 1 (FBLN1) (Fig. 5c and Extended Data Fig. 5e–g), allowing for colocalization and membrane interaction studies. FBLN1-expressing PI16⁺ RCs were located in the blood vessel-rich subepithelial zone (Fig. 5a,b,d and Extended Data Fig. 5a,b), which was densely populated by CD3⁺ T cells and CD20⁺ B cells (Fig. 5e and Extended Data Fig. 5h,j). Visualization of surface contact areas facilitated the demarcation of abundant membrane contacts between PI16⁺ RCs and CD20⁺ B cells (Fig. 5f and Extended Data Fig. 5i,k, arrows) and CD3⁺ T cells (Fig. 5f and Extended Data Fig. 5i,k, arrowheads). In sum, these data reveal a distinct PI16⁺ RC subset that is present in adult palatine tonsils and forms a niche adjacent to the lymphoreticular epithelium.

## PI16⁺ RC-mediated modulation of T cell activity

To characterize the interaction of FRC subsets with immune cells in more detail, we performed extended gene set enrichment and pathway analyses to determine the age- and inflammation-associated changes in TRCs (Extended Data Fig. 6a,b) in comparison to alterations in *PI16*⁺ RCs induced by tissue inflammation (Fig. 6a). We found minor age-dependent changes within TRCs (Extended Data Fig. 6a), whereas inflammation was associated with pronounced upregulation of distinct gene sets (Extended Data Fig. 6b). *PI16*⁺ RCs from the inflamed tonsils revealed a similarly strong induction of cellular activation processes during inflammation (Fig. 6a). The pro-angiogenic and niche-forming

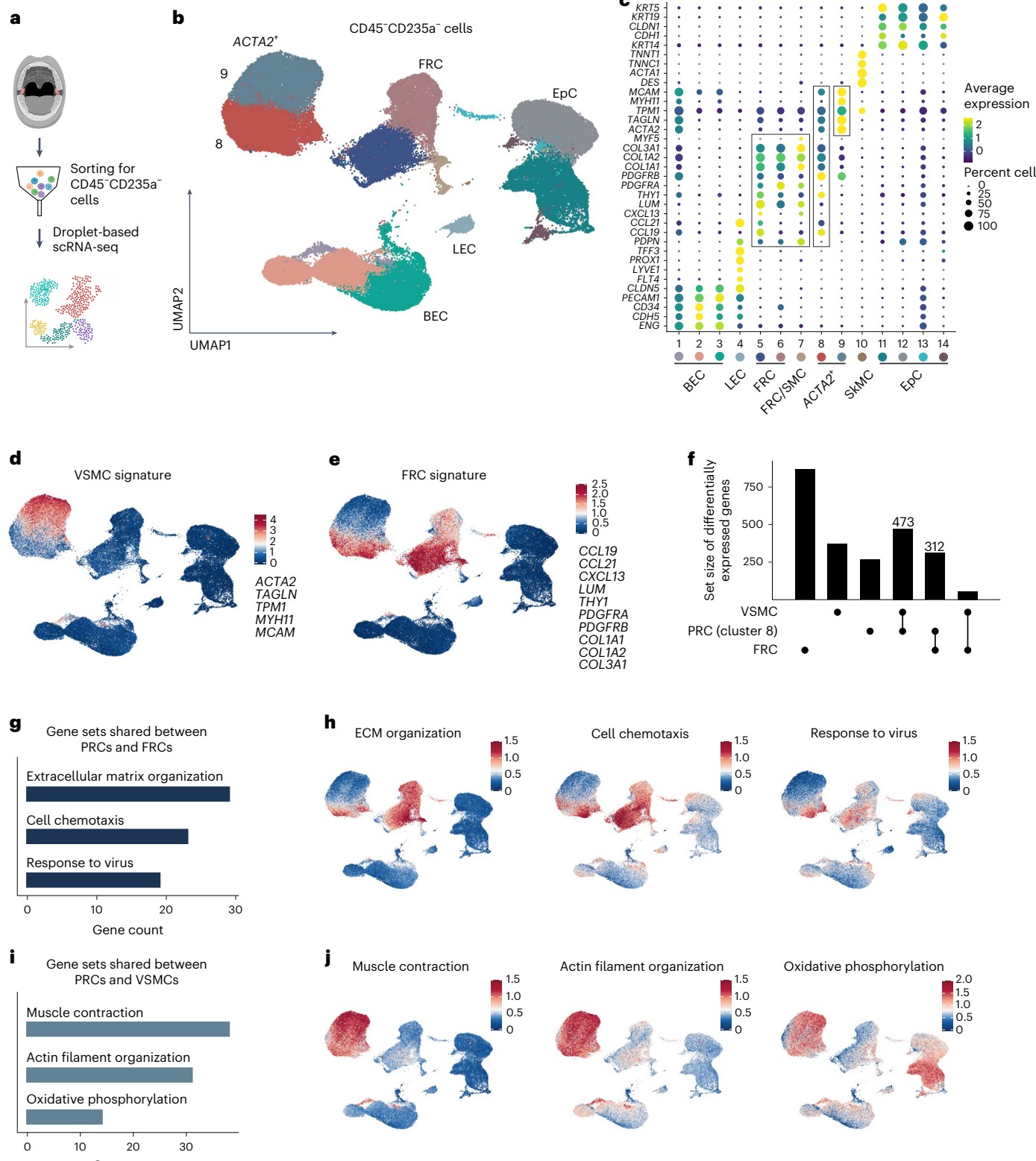

**Fig. 3 | Single-cell transcriptomic analysis of human palatine tonsil stromal cells. a**, Schematic representation of the scRNA-seq workflow. **b**, UMAP showing 14 CD45⁻CD235a⁻ stromal cell clusters after removal of contaminating cells (Methods) including BECs, LECs, epithelial cells, *ACTA2*⁺ cells and FRCs. **c**, Dot plot depicts marker genes used for characterization and assignment of the indicated stromal cell types. Frames highlight marker genes of FRC clusters and the two *ACTA2*⁺ clusters (8 and 9). SMC, smooth muscle cell; SkMC, skeletal muscle cell. **d,e**, Expression patterns of VSMC and FRC gene signatures projected onto UMAPs. **f**, UpSet plot showing differentially expressed genes shared between VSMCs, PRCs in cluster 8 and FRCs. **g,i**, Top significantly enriched terms according to gene ontology (GO) enrichment analysis based on differentially expressed genes shared between the indicated cell types. Bar plots show counts of genes assigned to respective cellular processes. **h,j**, Expression patterns of genes assigned to the indicated cellular processes projected onto UMAPs. scRNA-seq data represent a total of 86,966 CD45⁻CD235a⁻ stromal cells from *n* = 12 patients. ECM, extracellular matrix.

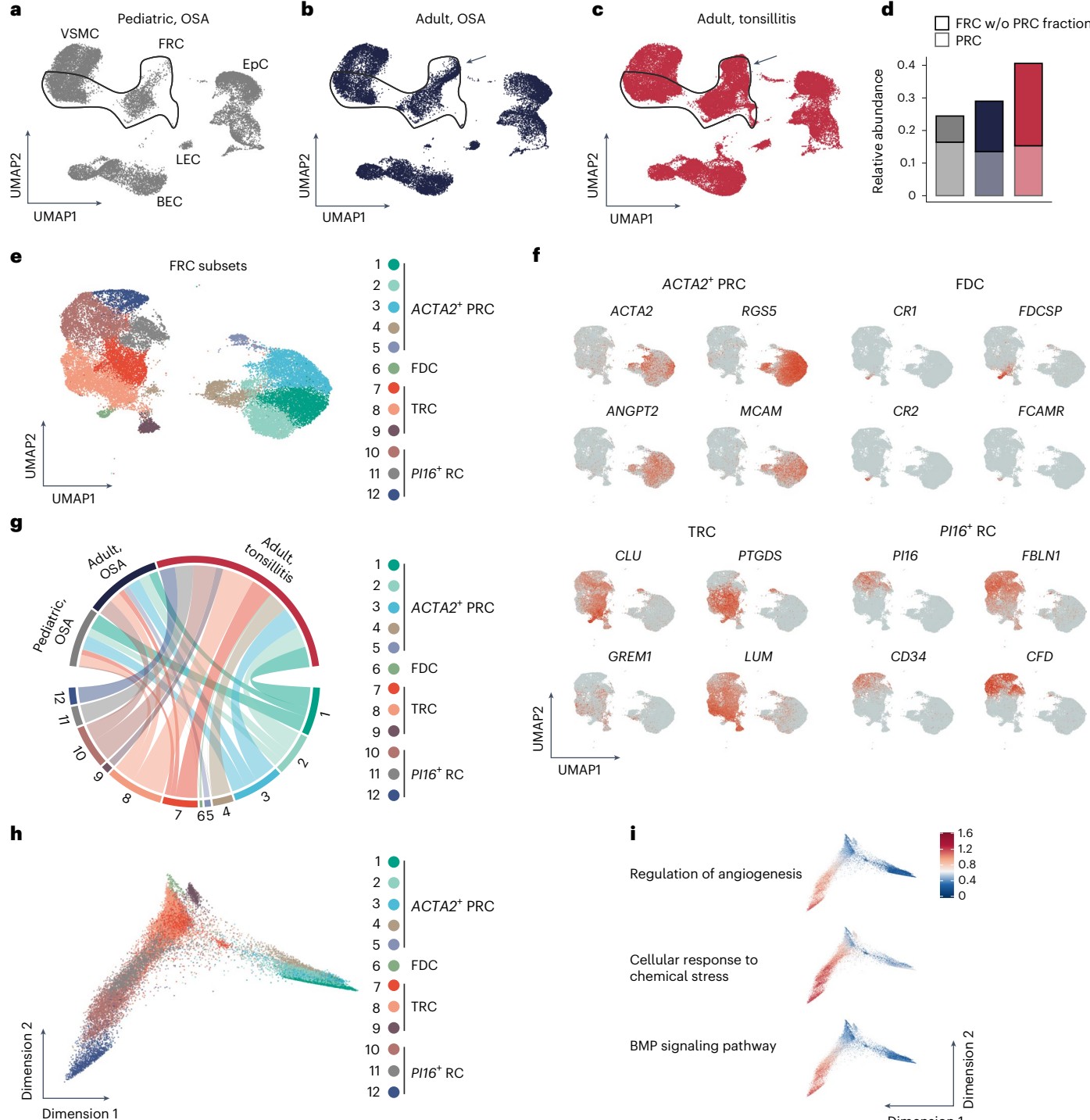

**Fig. 4 | Age- and inflammation-related molecular changes in tonsillar FRCs.**
**a**–**c**, UMAPs display scRNA-seq data of tonsillar CD45⁻CD235a⁻ stromal cells separated and colored according to different conditions. FRCs showing strong transcriptional changes are highlighted by arrows. **d**, Bar plot shows the relative abundance of FRCs among CD45⁻CD235a⁻ stromal cells according to different conditions as revealed by scRNA-seq. w/o, without. **e**, UMAP visualization of re-embedded FRC subsets. FDC, follicular dendritic cells. **f**, Feature UMAPs show expression pattern of cluster marker genes used for characterization of the indicated FRC subsets. **g**, Chord diagram shows the proportion of cells derived from different conditions for each FRC subset. **h**, Diffusion map dimensionality reduction of FRC subsets. **i**, Top significantly enriched GO terms in *PI16*⁺ RCs according to enrichment analysis based on subset marker genes. Expression pattern of genes assigned to the indicated cellular processes is projected onto diffusion maps. BMP, bone morphogenic protein. scRNA-seq data represent a total of 86,966 CD45⁻CD235a⁻ stromal cells and 28,571 FRCs containing 20,158 cells (4,867 FRCs) from *n* = 3 pediatric patients with OSA, 21,596 cells (6,184 FRCs) from *n* = 5 adult patients with OSA and 45,212 cells (17,520 FRCs) from *n* = 4 adult patients with tonsillitis.

functions of *PI16*⁺ RCs were underscored by transcriptional changes in genes assigned to the gene sets including 'regulation of angiogenesis' and 'response to molecule of bacterial origin' (Fig. 6b).

To further assess stromal–immune cell interactions, we sorted T and B cells from tonsils of two adult patients with OSA and three adult patients with tonsillitis for single-cell transcriptomic analysis (Extended

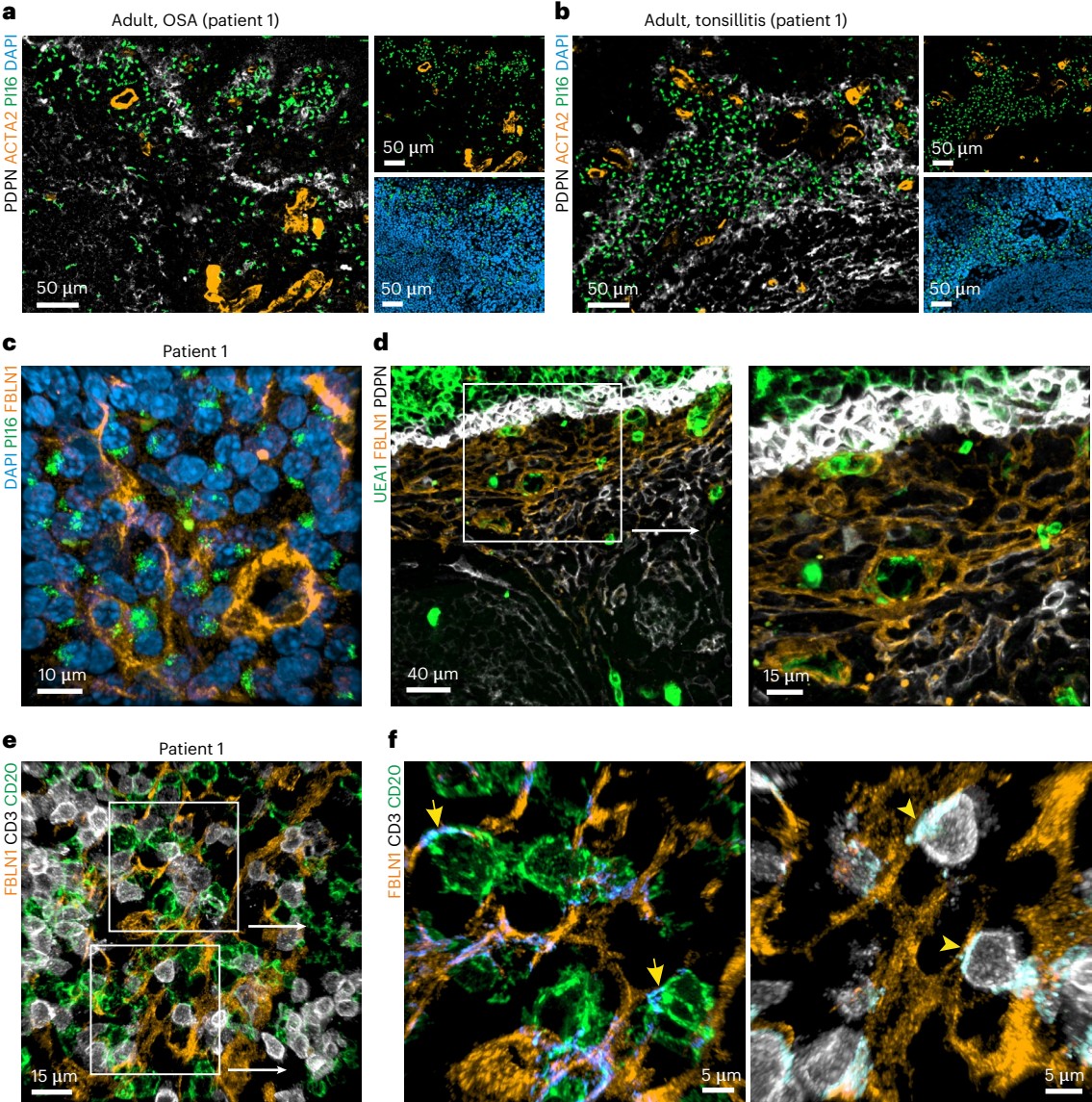

**Fig. 5 | Histological characterization of the PI16⁺ RC-underpinned subepithelial T and B cell niche. a,b,** PI16-positive cells in the subepithelial niche of tonsils from adult patients with OSA or tonsillitis. Cryosections were stained with the indicated antibodies and analyzed by confocal microscopy. **c,** High-resolution analysis of PI16⁺ cells shows intracellular PI16 signal in FBLN1⁺ reticular cells. Sections were stained with anti-FBLN1 and anti-PI16, and data were acquired by confocal microscopy and reconstructed in 3D. **d,** Morphology of the FBLN1⁺ subepithelial compartment in adult tonsils. Histological sections were stained with the indicated antibodies, and data were acquired and analyzed by confocal microscopy. The boxed area shows an FBLN1⁺ reticular network around UEA1⁺ endothelial cells at higher magnification on the right. **e,** High-resolution reconstruction of the adult subepithelial FBLN1⁺ FRC niche occupied by CD3⁺ and CD20⁺ lymphocytes. Sections were stained with the indicated antibodies, and data were acquired by confocal microscopy and reconstructed in 3D. **f,** Surface contact areas (blue) of FBLN1⁺ cells with CD20⁺ (arrows) and CD3⁺ (arrowheads) cells are shown at higher magnification. Microscopy images are representative for *n* = 3 adult patients with OSA and *n* = 3 adult patients with tonsillitis.

Data Fig. 6c and Extended Data Table 1). The merged scRNA-seq datasets allowed identification of the major T and B cell subsets according to established phenotypical and functional markers[30] (Extended Data Fig. 6d–g). Assessment of the aggregated cell–cell communication networks from all signaling pathways using the CellChat package[31] identified FRCs as highly interactive compared to T and B cells (Fig. 6c). Extended analysis of sender and receiver cell types suggested strong autocrine regulatory pathway activity by FRCs and predicted signaling from FRCs to T and B cells (Extended Data Fig. 7a,b) via pathways including collagen and fibronectin 1 (FN1) (Fig. 6d and Extended Data Fig. 7c). Moreover, interaction analysis predicted FRC stimulation by transforming growth factor (TGF)-β signals from several T and B cell subsets and autocrine fibroblast growth factor (FGF) signaling (Fig. 6e and Extended

Data Fig. 7c,d). Comparison of pathway networks between patients with OSA and tonsillitis across all FRC subsets indicated substantially increased intercellular communications during inflammation (Fig. 6f). Moreover, *PI16*⁺ RCs were among the dominant sender cells in the C–X–C motif chemokine ligand (CXCL) signaling pathway network (Extended Data Fig. 7e) and were predicted to be the strongest sender cells in the thrombospondin (THBS) signaling pathway network (Extended Data Fig. 7f). Further pro-inflammatory signals provided by *PI16*⁺ RCs to T and B cells included interleukin (IL)-6 and intercellular adhesion molecule 1 (ICAM1) (Extended Data Fig. 7g,h). Tonsillar FRCs and, in particular, *PI16*⁺ RCs integrate reciprocal signals from T and B cells such as LIGHT (tumor necrosis factor superfamily member 14 (TNFSF14)) through the lymphotoxin-β receptor (LTBR, TNFRSF3) (Extended Data Fig. 7i).

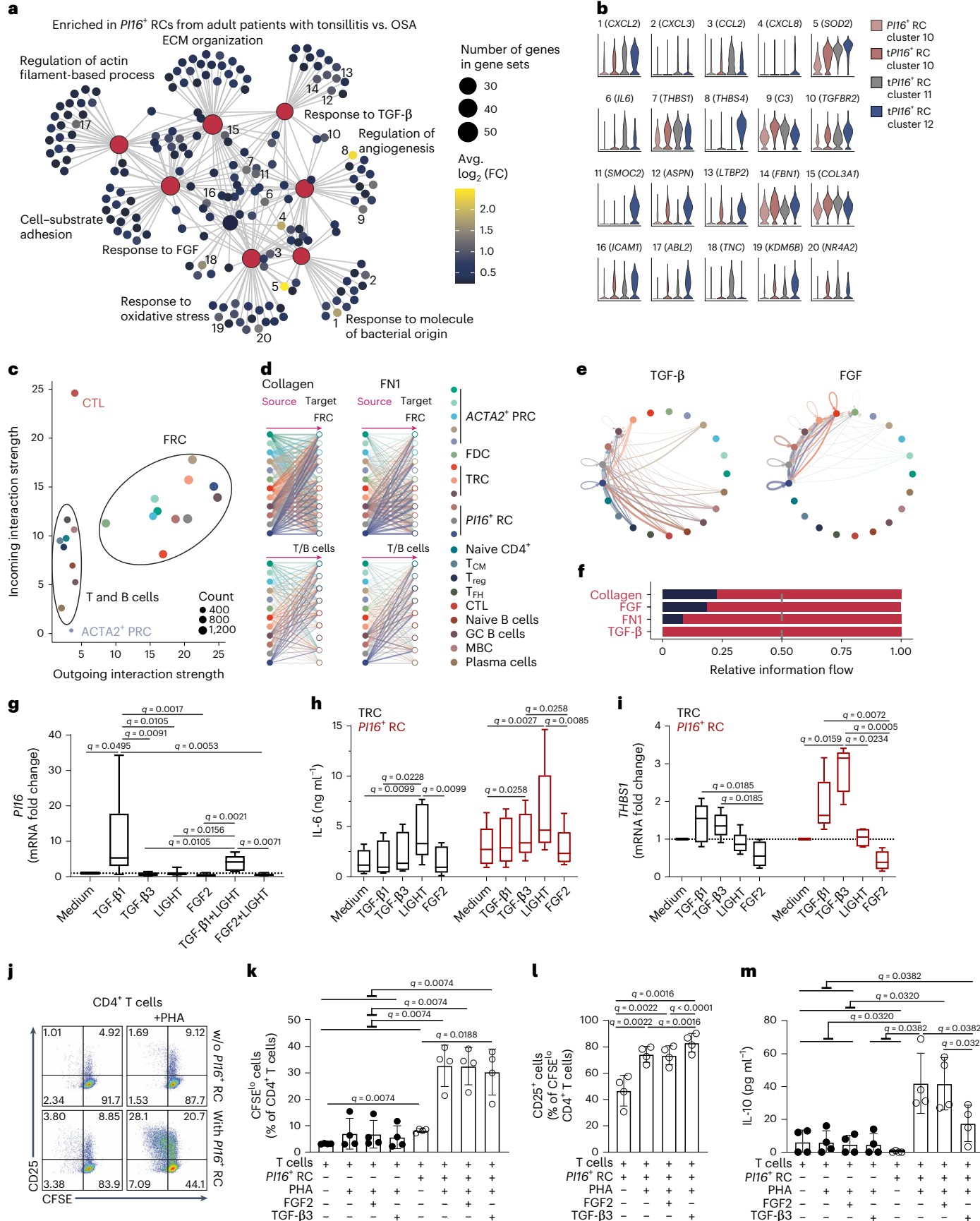

To validate whether and to what extent the predicted ligands or stimulatory factors affect tonsillar FRCs, we first expanded bulk tonsillar fibroblasts and stimulated the cells in vitro with recombinant human TGF-β1, TGF-β3, LIGHT and FGF2 following protocols established by Mourcin et al.[32] and Bar-Ephraim et al.[33]. We found that TGF-β1 strongly affects *PI16*+ RC differentiation as revealed by *PI16* mRNA expression in

**Fig. 6 | Interactome analysis of *PI16*+ RCs during homeostasis and inflammation. a**, Gene sets enriched upon inflammation based on differentially expressed genes in tonsillitis (t)*PI16*+ RCs compared to *PI16*+ RCs of patients with OSA. Numbers highlight top genes with an average log$_2$ (fold change) (avg. log$_2$ (FC)) > 1 assigned to enriched terms. **b**, Violin plots show gene expression profiles in different *PI16*+ RC clusters. **c**, Scatterplot indicates interaction strengths of individual cell subsets based on all signaling pathways derived. Dot size reflects interaction count. **d**, Hierarchical plots show inferred interactions of collagen and FN1 signaling networks. **e**, Circle plots represent inferred TGF-β and FGF signaling networks. **d**,**e**, Edge width reflects the communication probability. **f**, Significantly different (paired Wilcoxon test) relative information flow of signaling pathways between adult patients with OSA and tonsillitis. **g**–**i**, Bulk tonsillar fibroblasts from *n* = 8 (medium, TGF-β1, LIGHT), *n* = 6 (FGF2, TGF-β1 + LIGHT) and *n* = 5 (TGF-β3, FGF2 + LIGHT) patients with OSA (**g**) or sorted TRCs and *PI16*+ RCs from *n* = 5 adult patients with OSA or tonsillitis (**h**,**i**) were stimulated with recombinant proteins for 48 h. *PI16* (**g**) and *THBS1* (**i**) mRNA fold change measured by quantitative PCR with reverse transcription (RT−qPCR).

**h**, IL-6 concentration in culture supernatants. **j**–**m**, Carboxyfluorescein succinimidyl ester (CFSE)-labeled T cells were cultured (3 d) with *PI16*+ RCs and recombinant proteins and were stimulated with PHA. **j**, Representative flow cytometric plots depicting CD25 expression on CFSE$^{lo}$CD4+ T cells. **k**, Quantification of CFSE$^{lo}$ cells as a percentage of CD4+ cells. **l**, Quantification of CD25+ cells as a percentage of CFSE$^{lo}$CD4+ cells. **m**, IL-10 concentration in culture supernatant. **g**–**i**, Box extends from 25th to 75th percentiles, median is indicated and whiskers show the minimum and maximum values. **k**–**m**, Mean and s.d. are indicated, *n* = 4 patients with OSA or tonsillitis. *q* values were calculated with the Kruskal−Wallis (**g**,**i**), Friedman (**h**) or repeated measure (RM) one-way ANOVA (**k**–**m**) test and Benjamini, Krieger and Yekutieli correction for multiple comparisons. Interactome analysis represents scRNA-seq data of 81,997 T and B cells from *n* = 2 adult patients with OSA and *n* = 3 adult patients with tonsillitis and 23,704 FRCs from *n* = 5 adult patients with OSA and *n* = 4 adult patients with tonsillitis. CTL, cytotoxic T lymphocytes; T$_{CM}$, central memory T cells; T$_{reg}$, regulatory T cells; T$_{FH}$, follicular helper T cells; MBC, memory B cells.

tonsillar fibroblasts (Fig. 6g). Likewise, TGF-β1 stimulation increased expression of *FN1*, encoding a factor that was predicted to be upregulated in *PI16*+ RCs during inflammation (Extended Data Fig. 8a). TGF-β1 and LIGHT fostered IL-6 secretion by tonsillar fibroblasts, whereas FGF2 decreased IL-6 production (Extended Data Fig. 8b). In addition, LIGHT induced upregulation of ICAM1 surface expression, whereas FGF2 reduced ICAM1 expression (Extended Data Fig. 8c). Stimulation with both TGF-β1 and LIGHT showed that these factors balance the inflammatory reaction of tonsillar fibroblasts (Extended Data Fig. 8c). Of note, FGF2 appears to antagonize the effects of LIGHT on IL-6 production and ICAM1 expression (Extended Data Fig. 8b,c). Overall, these data confirm that TGF-β, FGF and LIGHT regulate the phenotype and function of tonsillar fibroblasts.

Next, we sorted *PI16*+ RCs and TRCs as a control population from adult patients with OSA and tonsillitis (Extended Data Fig. 8d) and stimulated the in vitro cultured FRC subsets with TGF-β1, TGF-β3, LIGHT or FGF2. We found that FGF2 serves as a proliferation factor for TRCs and *PI16*+ RCs (Extended Data Fig. 8f). LIGHT stimulation yielded activation of both TRCs and *PI16*+ RCs, leading to increased ICAM1 surface expression (Extended Data Fig. 8g) and IL-6 production (Fig. 6h). By contrast, TGF-β3-dependent stimulatory effects on IL-6 production appeared to be restricted to *PI16*+ RCs (Fig. 6h). Moreover, both TGF-β1 and TGF-β3 induced *THBS1* expression mainly in *PI16*+ RCs as predicted by the bioinformatic analysis (Fig. 6i). In sum, the interactome analysis in combination with in vitro validation of selected molecular circuits revealed *PI16*+ RCs as highly interactive cells that integrate autocrine and paracrine signals from both FRCs and immune cells to form a specific niche environment.

To assess to what extent *PI16*+ RCs modulate T cell activation and whether *PI16*+ RC-derived factors affect T cell functionality, tonsillar *PI16*+ RCs from patients with OSA or tonsillitis were co-cultured with autologous CD3+ T cells in the presence or absence of the surface molecule-cross-linking lectin phytohemagglutinin (PHA). Significant proliferation of tonsillar CD4+ T cells (Fig. 6j,k) or CD8+ T cells (Extended Data Fig. 9a,b) was only detectable in the presence of *PI16*+ RCs when compared to that with PHA stimulation alone. PHA-mediated activation of T cells in the presence of *PI16*+ RCs resulted in significantly increased expression of the T cell-activation marker CD25 by CD4+ T cells (Fig. 6j,l) and CD8+ T cells (Extended Data Fig. 9a,c). TGF-β3 enhanced T cell activation (Fig. 6l and Extended Data Fig. 9c,d) and significantly decreased IL-10 production by T cells (Fig. 6m). At the same time, PI16+ RCs maintained production of IL-6 in the presence of FGF2 or TGF-β3 in the co-culture system (Extended Data Fig. 9e), supporting the notion that *PI16*+ RCs employ discrete signaling pathways to support T cell activation and differentiation.

## Discussion

Palatine tonsils play an important role in the initiation and control of innate and adaptive immune responses in the human oropharynx because both oral and air-borne pathogens have to pass this particular region before being conveyed into the gastrointestinal and/or respiratory tract. The first key function of any SLO is the capture of antigenic material and transfer of native or processed antigens to those regions that harbor lymphocytes[3,9]. The antigen-capturing zone of SLOs is usually underpinned by marginal reticular cells (MRCs), for example, in the marginal sinus of the spleen[6] or the subcapsular sinus of lymph nodes[34,35]. Here, we found that MRCs are not represented in the FRC landscape of human palatine tonsils, which is in line with a previous immunohistological marker-based analysis[32]. It is conceivable that the key SLO function of antigen sampling is entirely covered by the lymphoreticular epithelium that forms a membranous meshwork intermingled with lymphocytes and myeloid cells. Indeed, previously described (M) cell markers of adenoids[36] and palatine tonsils such as class II β-tubulin[37] and clusterin[38] are broadly expressed across epithelial cells in our transcriptomic analysis, whereas none of the cell clusters revealed a specific M cell signature. Thus, antigenic material present in the oropharynx is most likely either passively pressed through the membranous lymphoreticular crypt epithelium as a liquid condensate or actively transported further into the lymphoid tissue by myeloid cells that are abundant in the specialized epithelial barrier.

A second crucial compartment that determines the functionality of SLOs comprises the blood and lymphatic vasculature, which together regulate immune cell influx and efflux and maintain the liquid balance of the tissue[9]. Human palatine tonsils are highly vascularized as shown here by histological and flow cytometric analyses and as exemplified by the high proportion (20–25%) of BECs among all non-hematopoietic cells in the transcriptomic analysis. Larger blood vessels in lymphoid organs such as the spleen are surrounded by perivascular fibroblastic stromal cells that are arranged in several concentric layers with fibroblastic mural cells directly underlying the blood vessel endothelium[6]. Further layers around blood vessels consist of PRCs, which connect the perivascular compartment with specialized immune cell niches such as the T cell zone in the murine spleen[6] and Peyer's patch[7] and form the perivascular fraction of the conduit system in murine lymph nodes[39]. The perivascular fibroblastic stromal cells of the human palatine tonsil shared the stereotypic organization described for murine SLOs with VSCMs (that is, one specific fraction of mural cells[24,40]), forming the first concentric layers and PRCs bridging the space to T and/or B cell-rich areas. It will be important in future studies to resolve the identity and relation of PRCs and VSCMs in other human SLOs.

It has been suggested that *PI16*+ fibroblasts of different tissues and tumors coexpressing *CD34*, *HAS1* and *PLIN2* represent an adventitial fibroblast subset that may serve as a 'universal' progenitor of fibroblast lineages[29]. Although the transcriptomic profile of PI16+ RCs matches the key transcriptomic features of 'universal' *PI16*+ fibroblasts, we neither found this particular population in the PRC fraction of human palatine tonsils nor did PI16+ RCs appear as a prominent tonsillar FRC subset in pediatric patients with OSA. The peptidase inhibitor PI16 has been shown to promote transendothelial migration of leukocytes[41] and to inhibit the extracellular matrix-cleaving enzyme matrix metallopeptidase 2 (ref. [42]). The high diversity of immune-activating and extracellular matrix-remodeling pathways employed by PI16+ RCs reflects the ability of this FRC subset to interact with several immune cell types in distinct regions of SLOs. The specific cellular composition and activity of immune cells in SLOs may therefore result in imprinting of the PI16+ RC phenotype that differs from that of *PI16*-expressing fibroblasts in non-lymphoid tissues and tumors.

The almost complete absence of PI16+ RCs in pediatric tonsils indicates that this subepithelial FRC subset is formed mainly in response to continued exposure to antigenic material that flows into the lymphoid tissue through the lymphoreticular epithelium. Our results indicate that PI16+ RCs use canonical immune cell-stimulating pathways such as the pro-inflammatory cytokine IL-6 (ref. [43]) or the co-stimulatory adhesion molecule ICAM1 (ref. [44]) to increase immune cell activation. Moreover, our findings from the co-culture system of tonsillar PI16+ RCs with autologous T cells further support the notion that PI16+ RCs can foster T cell activation and differentiation. The cross-talk between FRCs and immune cells involves imprinting of FRC subset identity and modulation of FRC function to secure optimal molecular catering to immune cells. Our data suggest that LIGHT and TGF-β3 could serve as hitherto unknown secondary signals that drive FRC subset specification and activation under inflammatory conditions[18]. The involvement of these pathways in the formation of PI16+ RC-generated niches is in line with a cross-organ and cross-species comparison of immune cell-derived niche factors that determine the identity and function of CXCL13-expressing PI16+ RCs[45].

In sum, our study shows that PI16+ RCs underpin an adaptive subepithelial niche in mucosal lymphoid tissues of the human oropharynx and shape the microenvironment for efficient T cell activation and differentiation.

## Online content

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

## Methods

### Study patients and sample collection

The study protocol has been reviewed and approved by the Ethikkommission Ostschweiz (EKOS), permission numbers 2017-00051 (adult patients) and 2018-01646 (pediatric patients). All study participants provided written informed consent in accordance with the Declaration of Helsinki and the International Conference on Harmonization Guidelines for Good Clinical Practice. All regulations were followed according to the Swiss authorities and according to the clinical protocols. Sample size calculation for the study was not performed because of the exploratory nature of the study design. Tonsil samples were collected from adult and pediatric patients suffering from OSA or tonsillitis that underwent routine tonsillectomy (Extended Data Table 1). After surgical excision, tonsils were collected in sterile phosphate-buffered saline (PBS) on ice and processed within 1 h of resection. For tonsils that were acquired at the University of Pennsylvania and included in the extended cohort, tonsil use was approved by a material-transfer agreement between the University of Pennsylvania and Children's Hospital of Philadelphia (ID 58590/00). These tonsils were received without any protected health information or identifiers, exempt from review by the Children's Hospital of Philadelphia Institutional Review Board (SOP 407, section XII 'Secondary Use of De-Identified Data or Specimens').

### Isolation of stromal and immune cells from palatine tonsils

Heavily clotted or cauterized areas of the tissue were removed, and tonsils were dissected into small pieces. For the isolation of stromal cells, tissue was digested using a MACS human tumor dissociation kit (Miltenyi Biotec) according to the manufacturer's protocol. To enrich for stromal cells, hematopoietic and erythrocytic cells were depleted by incubating the cell suspension with MACS anti-CD45 and anti-CD235a (glycophorin A) MicroBeads (Miltenyi Biotec) and passing it through a MACS LS column (Miltenyi Biotec). The unbound single-cell suspension was stained for further flow cytometric analysis or cell sorting. For the isolation of hematopoietic cells, tissues were gently smashed across a 26-gauge wire mesh using a syringe plunger and washed with medium (RMPI 1640 containing 2% fetal calf serum (FCS; Sigma), 1% penicillin–streptomycin (Sigma) and 20 mM HEPES (Sigma)) until further staining for flow cytometric analysis or cell sorting.

### Antibody staining and flow cytometric analysis

Single-cell suspensions of tonsillar stromal cells were prepared as described above, incubated with Fixable Viability Stain 510 (1:1,000, BD Biosciences) and subsequently stained for 20 min at 4 °C in PBS containing 2% FCS and 2 mM EDTA with the FITC-conjugated lectin UEA1 (1:100, Sigma) and the following fluorochrome-conjugated antibodies: anti-human PDPN, anti-human CD45, anti-human CD235a and anti-human CD31. Next, intracellular staining was performed by treatment with FoxP3 Transcription Factor Staining Buffer (eBioscience) for 30 min, and subsequent staining was carried out with anti-human ACTA2 in permeabilization buffer as indicated in the manufacturer's protocol. Cells were washed with permeabilization buffer, resuspended in PBS containing 2% FCS and 2 mM EDTA and acquired with a BD LSR-Fortessa and FACSDiva (BD Biosciences, version 8.0.1 and version 9.0.1) software. Analysis was performed using FlowJo software version 10.6.2 (Tree Star) and the PhenoGraph (version 3.0) clustering algorithm. GraphPad Prism (version 8) was used for statistical analyses. Differences with a *P* value <0.05 were considered statistically significant.

### Cell sorting and droplet-based single-cell RNA-sequencing analysis

Single-cell suspensions of stromal and hematopoietic cells were prepared as described above. For in vitro differentiation assays, stromal cells were stained with fluorochrome-conjugated antibodies to human CD45, CD235a, PDPN and CD34, a biotin-conjugated antibody to CD31 and biotin-conjugated lectin UEA1 (1:100, Sigma),

which were both detected with APC-Cy7-conjugated streptavidin (BioLegend, 405208, 1:500). Hematopoietic cells were stained with fluorochrome-conjugated antibodies to human CD45, CD19 and CD3. Live–dead cell discrimination was performed by adding 7-aminoactinomycin D (7AAD; Calbiochem) before acquisition. CD45⁻CD235a⁻UEA1⁻CD31⁻PDPN⁺CD34⁻ TRCs, CD45⁻CD235a⁻UEA1⁻CD31⁻PDPN⁺CD34⁺*PI16*⁺ RCs and CD45⁺CD19⁻CD3⁺ T cells were sorted with a BD FACSMelody cell sorter and FACSChorus (BD Biosciences, version 1.3) software. For scRNA-seq analyses, hematopoietic cells were stained with fluorochrome-conjugated antibodies to human CD45, CD3, CD19 and CD14 and stromal cells were stained with fluorochrome-conjugated antibodies to human CD45 and CD235a. Live–dead cell discrimination was performed by adding 7AAD (Calbiochem) before acquisition. CD45⁻CD235a⁻ stromal cells, CD45⁺CD3⁺ T cells and CD45⁺CD19⁺ B cells were sorted with a BD FACSMelody cell sorter and FACSChorus (BD Biosciences, version 1.3) software and run on the 10x Chromium analyzer (10x Genomics)[46]. cDNA library generation was performed following the established commercial protocol for the Chromium Single Cell 3′ Reagent Kit (version 3 chemistry). Libraries were sequenced using a NovaSeq 6000 sequencer at the Functional Genomic Center Zurich. Samples from 12 patients were collected at KSSG as indicated in Extended Data Table 1 and processed in nine batches. Samples from three patients were collected at the University of Pennsylvania (Extended Data Table 1). Gene expression estimation from sequencing files was carried out using CellRanger (version 3.0.2)[47] count with the Ensembl GRCh38.9 release as the reference to build the index for human samples. Next, quality control was performed in R version 4.0.0 using the R–Bioconductor package scater (version 1.16.0)[48] and included removal of damaged and contaminating cells based on (1) very high or low UMI counts (>2.5 median absolute deviation from the median across all cells), (2) very high or low total number of detected genes (>2.5 median absolute deviation from the median across all cells) and (3) high mitochondrial gene content (>2.5 median absolute deviations above the median across all cells). In addition, contaminating cells expressing any of the markers *CD3E*, *PTPRC*, *CD79A* or *GYPA* were removed from stromal cell samples. Following quality control, the dataset included 87,830 stromal cells (pediatric OSA, 20,240 cells; adult OSA, 21,688 cells; adult tonsillitis, 45,902 cells) and 82,092 immune cells (adult OSA, 27,979 cells; adult tonsillitis, 54,113 cells).

### Comparative analysis and interactome analysis

Downstream analysis was performed using the Seurat R package (version 4.0.1)[49,50] and included normalization, scaling, dimensionality reduction with PCA and UMAP, graph-based clustering and calculation of unbiased cluster markers as well as dimensionality reduction with diffusionmap as implemented in the scater R–Bioconductor package (version 1.16.0)[48]. Clusters were characterized based on expression of calculated cluster markers and canonical marker genes as reported in previous publications[17,16,24,25]. For the extended stromal cell analysis, two contaminating clusters with 564 cycling cells and 300 cells expressing both fibroblast and endothelial marker genes (indicative of doublets) were removed. For high-resolution FRC analysis, FRC subsets were re-embedded, and three clusters containing 967 cells with endothelial, neuronal or smooth muscle cell genes were excluded. Comparative analysis included determination of cell type-, subset- and condition-specific gene signatures. Thereby differentially expressed genes were calculated by running the FindAllMarkers function from the Seurat R package. Functional signatures were further derived by running GO enrichment analysis on subset-specific or condition-specific genes using the clusterProfiler R–Bioconductor package (version 3.15.3)[51]. Integration was performed using the IntegrateData function from the Seurat package.

For interactome analysis, all acquired immune cell datasets were first merged and characterized using normalization, scaling, UMAP dimensionality reduction, graph-based clustering and calculation of

of unbiased cluster markers. Clusters were characterized based on expression of calculated cluster markers and canonical marker genes as reported previously[30], and one cluster containing 95 myeloid cells was removed. CellChat (version 1.1.3)[31] was used as a tool to infer ligand–receptor interactions based on scRNA-seq data. CellChat applies a signaling molecule interaction database (CellChatDB) to predict intercellular communication patterns based on differentially overexpressed receptors and ligands. Known ligand–receptor complexes based on the KEGG Pathway database are used by the software to compute the communication probability on a pathway level. Furthermore, complex cell–cell communication networks are analyzed using centrality measures from network graphs to identify major signaling sources, targets, influencers and mediators. Here, cells from adult patients with OSA and adult patients with tonsillitis were analyzed collectively for the identification of predominant interaction patterns and individually to determine enriched interactions within a certain condition.

## Immunohistochemistry and microscopy
Tissues were processed either for vibratome or cryotome sectioning. For vibratome processing, tissues were fixed at 4 °C overnight with 4% paraformaldehyde (Merck Millipore) in PBS with agitation. Fixed tissues were further washed with PBS containing 1% Triton X-100 (Sigma) and 2% FCS (Sigma) overnight at 4 °C. Tissues were embedded in 4% NuSieve GTG low melting agarose (Lonza) in PBS, and 40-μm sections were generated with a vibratome (Leica VT1200). For cryotome sectioning, tissues were embedded in FSC 22 Clear (Leica Biosystems), frozen fresh in an isopropanol–dry ice bath and stored at −80 °C. Sections (10 μm) generated with a cryostat (Leica CM1950) were mounted onto Thermo slides and fixed for 10 min in methanol at −20 °C. Tissues were blocked in PBS containing 10% FCS, 1 mg ml⁻¹ anti-Fcγ receptor (BD Biosciences) and 0.1% Triton X-100 (Sigma) at 4 °C for 2 h.

Sections were further incubated overnight with FITC-conjugated lectin UEA1 (1:500, Sigma) and the following labeled antibodies: anti-human ACTA2–Cy3, anti-human CD324–APC, anti-human CD20–A488 and anti-human CD11c–APC. The unconjugated antibodies anti-human CD3, anti-human PDPN, anti-human FBLN1 and anti-human PI16 were detected with the following secondary antibodies: anti-rabbit IgG–A488, anti-rat IgG–A488, anti-mouse IgG–A488, anti-rat IgG–A594, anti-rat IgG–A647, anti-mouse IgG–A647, anti-rabbit IgG–Cy3, anti-rat IgG–Cy3 and anti-mouse IgG–Cy3. Microscopy was performed using a confocal microscope (LSM 980, Carl Zeiss), and images were recorded and processed with ZEN software (Carl Zeiss, version 3.3). Imaris version 9 (Bitplane) was used for image analysis and for rendering, masking, reconstruction and histomorphometric and volumetric analysis of confocal z stacks to histologically assess cell surface contact areas and intracellular PI16 localization. PI16 staining in cryosections was masked to highlight intracellular signal with a voxel count >200. Fluorescence intensities were quantified using ImageJ 1.49v software (Wayne Rasband).

## In vitro FRC differentiation and co-culture with T cells
For in vitro differentiation assays, bulk primary tonsillar fibroblasts and sorted TRCs and *PI16*⁺ RCs were cultured for 10–20 d in RPMI supplemented with 10% FCS (Sigma), 1% penicillin–streptomycin (Sigma) and 16 μg ml⁻¹ gentamicin following protocols established by Mourcin et al.[32] and Bar-Ephraim et al.[33]. To evaluate differentiation and activation potential, cells were plated at 15,000 cells per cm² and stimulated for 48 h with recombinant human (rh)TGF-β1 (R&D Systems, 7754-BH/CF, 1 ng ml⁻¹), rhTGF-β3 (R&D Systems, 8420-B3/CF, 0.1 ng ml⁻¹), rhLIGHT (TNFSF14; R&D Systems, 664-LI/CF, 10 ng ml⁻¹), rhFGF2 (R&D Systems, 3718-FB, 10 ng ml⁻¹) or a combination of rhTGF-β1 plus rhLIGHT or rhFGF2 plus rhLIGHT. Cells were incubated with a BV421-conjugated antibody to human ICAM1 (CD54) and Fixable Viability Stain 510 (1:1,000, BD Biosciences), and data were acquired with a BD LSRFortessa and FACSDiva (BD Biosciences, version 9.0.1)

software for cell counting and ICAM1 MFI analysis using FlowJo software version 10.6.2 (Tree Star).

For in vitro cell interaction experiments, tonsillar *PI16*⁺ RCs from adult patients with OSA and tonsillitis were plated at 30,000 cells per cm² and co-cultured for 3 d with sorted, CellTrace CFSE dye-labeled tonsillar T cells (0.5 μM, Invitrogen) in the presence or absence of PHA (1 μg ml⁻¹, Sigma), rhTGF-β3 (R&D Systems, 8420-B3/CF, 0.1 ng ml⁻¹) and rhFGF2 (R&D Systems, 3718-FB, 10 ng ml⁻¹) in RPMI supplemented with 10% FCS (Sigma), 1% penicillin–streptomycin (Sigma), 16 μg ml⁻¹ gentamicin, rhIL-2 (75 U), 0.1% 2-mercaptoethanol (Gibco), 1% MEM non-essential amino acid solution (Sigma), 1% L-glutamine (Gibco) and 1% sodium pyruvate solution (Sigma). Cells were incubated with the indicated antibodies and Fixable Viability Stain 510 (1:1,000, BD Biosciences), and data were acquired with a BD FACSymphony A3 cell analyzer and FACSDiva (BD Biosciences, version 9.0.1) software and analyzed with FlowJo software version 10.6.2 (Tree Star). IL-6 and IL-10 concentrations in the supernatant were assessed using the BD Cytometric Bead Array (BD Biosciences, 558276) according to the manufacturer's protocol. GraphPad Prism (version 8) was used for statistical analyses.

## Quantitative PCR with reverse transcription
For RT–qPCR, total cellular RNA was extracted from cultured and stimulated cells using the Quick-RNA Microprep Kit (Zymo Research, R1051) following the manufacturer's protocol. cDNA was prepared using the High-Capacity cDNA Reverse Transcription Kit (Applied Biosystems, 4368814), and RT–qPCR was performed using PowerUp SYBR Green Master Mix (Applied Biosystems, Thermo Fisher Scientific) on a QuantStudio 5 machine (Applied Biosystems, Thermo Fisher Scientific). Expression levels were measured by using the following primers: *PI16* (QT00007000, Qiagen), *FN1* (QT00038024, Qiagen), *THBS1* (QT00028497, Qiagen) and *GAPDH* (QT00079247, Qiagen). Relative mRNA expression was calculated by the comparative cycling threshold method (ΔΔCt method), using the expression of glyceraldehyde 3-phosphate dehydrogenase (GAPDH) for normalization. Fold change of relative mRNA expression compared to the medium control was calculated. GraphPad Prism (version 8) was used for statistical analyses.

## Antibodies for flow cytometry and cell sorting
Anti-human PDPN–PE (clone NZ-1.3, 12-9381-42, 1:100), anti-human CD45–PE-Cy7 (clone HI30, 25-0459-42, 1:100), anti-human CD31–biotin (clone WM59, 13-0319-82, 1:100) and anti-human or mouse ACTA2–eFluor 660 (clone 1A4, 50-9760-82, 1:500) were purchased from eBioscience. Anti-human CD31–PerCP (clone WM59, 303132, 1:50), anti-human CD235a–PE-Cy7 (clone HI264, 349112, 1:100), anti-human CD3–FITC (clone UCHT1, 300440, 1:100), anti-human CD14–PE-Cy7 (clone MSE2, 301813, 1:100), anti-human CD19–APC/Fire 750 (clone HIB19, 302258, 1:100), anti-human CD54 (ICAM1)–BV421 (clone HA58, 353132, 1:100) and anti-human CD34–FITC (clone 581, 343504, 1:100) were purchased from BioLegend. Anti-human CD45–APC-Cy7 (clone 2D1, 560178, 1:100), anti-human CD4–BUV395 (clone SK3, 563550, 1:200), anti-human CD8–BUV805 (clone SK1, 612889, 1:400) and anti-human CD25–BUV563 (clone 2A3, 612918, 1:400) were purchased from BD Biosciences.

## Antibodies for histology
Anti-human CD324 (E-cadherin)–APC (clone 67A4, 324107, 1:200) and anti-human CD11c–A647 (clone 301613, B208071, 1:200) were purchased from BioLegend. Anti-human CD3 (polyclonal, A0452, 1:200) was purchased from Dako. Anti-human PDPN (clone NZ-1.3, 14-9381-82, 1:200) and anti-human CD20–A488 (clone L26, 53-0202-80, 1:200) were purchased from eBioscience. Anti-human or mouse ACTA2 Cy3 (clone 1A4, C6198, 1:1,000) was purchased from Sigma. Anti-human FBLN1 (clone CL0337, AMAb906960, 1:125) was purchased from Atlas Antibodies. Anti-human PI16 (NBP1-92254, 1:125) was purchased from Novus Biologicals.

Unconjugated antibodies were stained with the following secondary antibodies: anti-rabbit IgG–A488 (711-547-003, 1:500), anti-rat IgG–A488 (712-545-150, 1:1,000), anti-mouse IgG–A488 (715-545-150, 1:1,000), anti-rat IgG–A594 (712-585-153, 1:1,000), anti-rat IgG–A647 (712-605-153, 1:1,000), anti-mouse IgG–A647 (715-605-150, 1:1,000), anti-rabbit IgG–Cy3 (711-165-152, 1:1,000), anti-rat IgG–Cy3 (712-167-003, 1:500) and anti-mouse IgG–Cy3 (715-165-150, 1:1,000), all purchased from Jackson ImmunoResearch.

## Reporting summary

Further information on research design is available in the Nature Portfolio Reporting Summary linked to this article.

## Data availability

The scRNA-seq data generated in this study have been deposited in the BioStudies database (https://www.ebi.ac.uk/biostudies/) and are available under accession code E-MTAB-11715. GRCh38.9 was used as a reference genome to build the indices. Processed data files can be downloaded from the figshare platform (https://figshare.com) at https://doi.org/10.6084/m9.figshare.21325737 and explored via an interactive browser at https://immbiosg.github.io/FRCdataExplorer/.

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

## Acknowledgements

We thank L. Büchler for excellent technical assistance and E. Camiolo and G. Wertheim for procuring tonsil samples from the Children's Hospital of Philadelphia. This study received financial support from the Swiss National Science Foundation (grants 177208 and 182583 to B.L.) and the Research Commission of the Kantonsspital St. Gallen (grant 19/07 to Y.S.). N.B.P. was supported by an Ambizione research grant from the Swiss National Science Foundation (grant 180011) and a Hans Peter Hofschneider Professorship from the Stiftung für Experimentelle Biomedizin. J.D.B. was supported by a Doris Duke Charitable Foundation's Physician Scientist Fellowship, an American Society of Transplantation and Cellular Therapy New Investigator Award and the NHLBI (T32-HL07439 to J.D.B.). I.M. was supported by the NIAID (R01-AI091627). The funders had no role in study design, data collection and analysis, decision to publish or preparation of the manuscript.

## Author contributions

B.L. designed the study, discussed data and wrote the paper; A.D.M. and Y.S. performed experiments, analyzed data and wrote the paper; M.L. analyzed and discussed data; N.C., H.-W.C., L.O., N.B.P. and J.D.B. performed experiments and discussed data; and I.M. and S.J.S. discussed data and provided patient material.

## Competing interests

H.-W.C., L.O., N.B.P. and B.L. are founders and shareholders of Stromal Therapeutics. L.O. and B.L. are members of the board of Stromal Therapeutics. All other authors declare no competing interests.

## Additional information

**Extended data** is available for this paper at https://doi.org/10.1038/s41590-023-01502-4.

**Correspondence and requests for materials** should be addressed to Burkhard Ludewig.

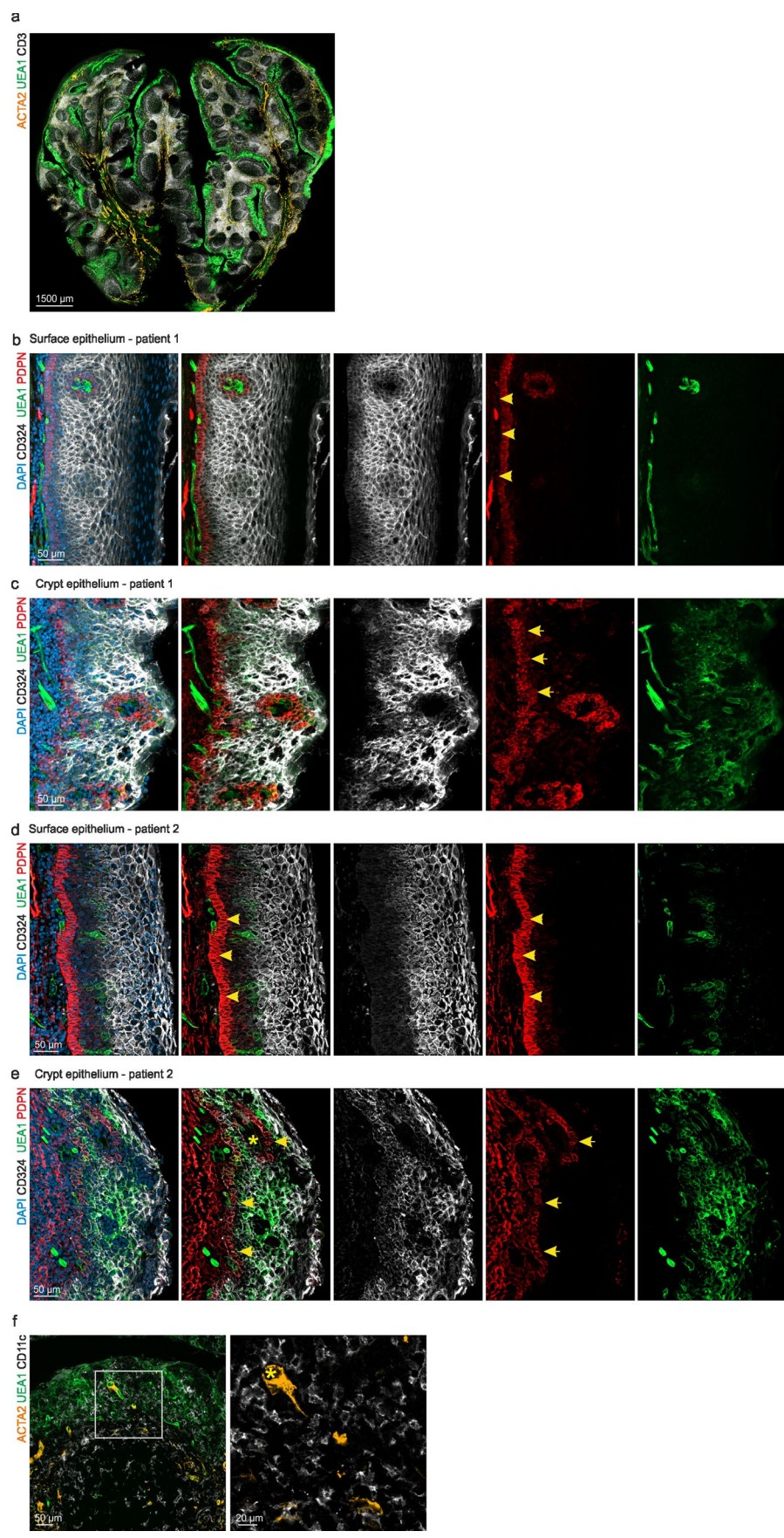

**Extended Data Fig. 1 | See next page for caption.**

**Extended Data Fig. 1 | Epithelial cell topology in human palatine tonsils. a-f**, Representative immunofluorescence images of palatine tonsil sections from adult patients with OSA stained with the indicated antibodies and analyzed by confocal microscopy. **a**, Structure of the palatine tonsil. UEA1 staining of the lymphoreticular epithelium highlights deep crypts permeating palatine tonsils. **b**,**d**, High resolution analysis of the non-keratinized stratified squamous surface epithelium. PDPN-expressing basal cuboidal epithelial cells are highlighted (arrowheads). **c**,**e**, Morphology of the lymphoreticular crypt epithelium. Broadened PDPN-expressing basal layer of the crypt epithelium is highlighted (arrows). **f**, Lymphoreticular crypt epithelium occupied by CD11c[+] myeloid cells. Boxed area shows magnified intraepithelial CD11c[+] cells surrounding a blood vessel (asterisk). Microscopy images are representative for $n$ = 3 adult patients with OSA.

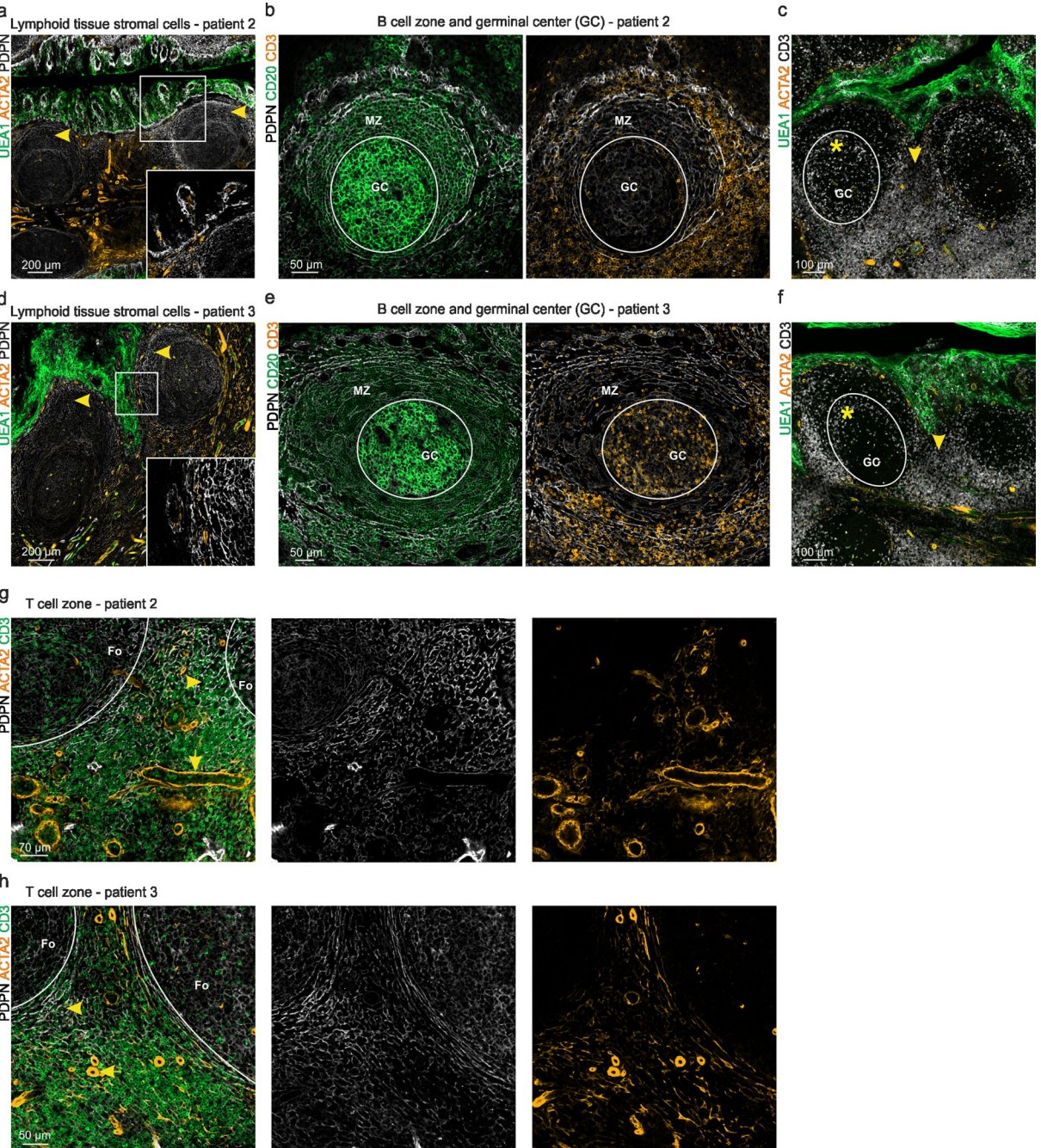

**Extended Data Fig. 2 | Lymphoid tissue stromal cell topology in human palatine tonsils. a–h**, Representative immunofluorescence images of palatine tonsil sections from adult patients with OSA stained with the indicated antibodies and analyzed by confocal microscopy. **a,d**, Epithelial and subepithelial areas (arrowheads) underpinned by PDPN+ stromal cells. Boxed area shows magnified epithelial-lymphoid tissue interface. **b,e**, B cell follicles with germinal centers (GC) and mantle zones (MZ) underpinned by PDPN+ FRCs. **c,f**, CD3+ lymphocytes in epithelial, germinal center (asterisk) and interfollicular T cell areas (arrowhead). **g,h**, T cell area surrounding B cell follicles (Fo) underpinned by PDPN+ FRCs (arrowhead) and ACTA2+ PDPN− VSMCs (arrow). Microscopy images are representative for n = 3 adult patients with OSA.

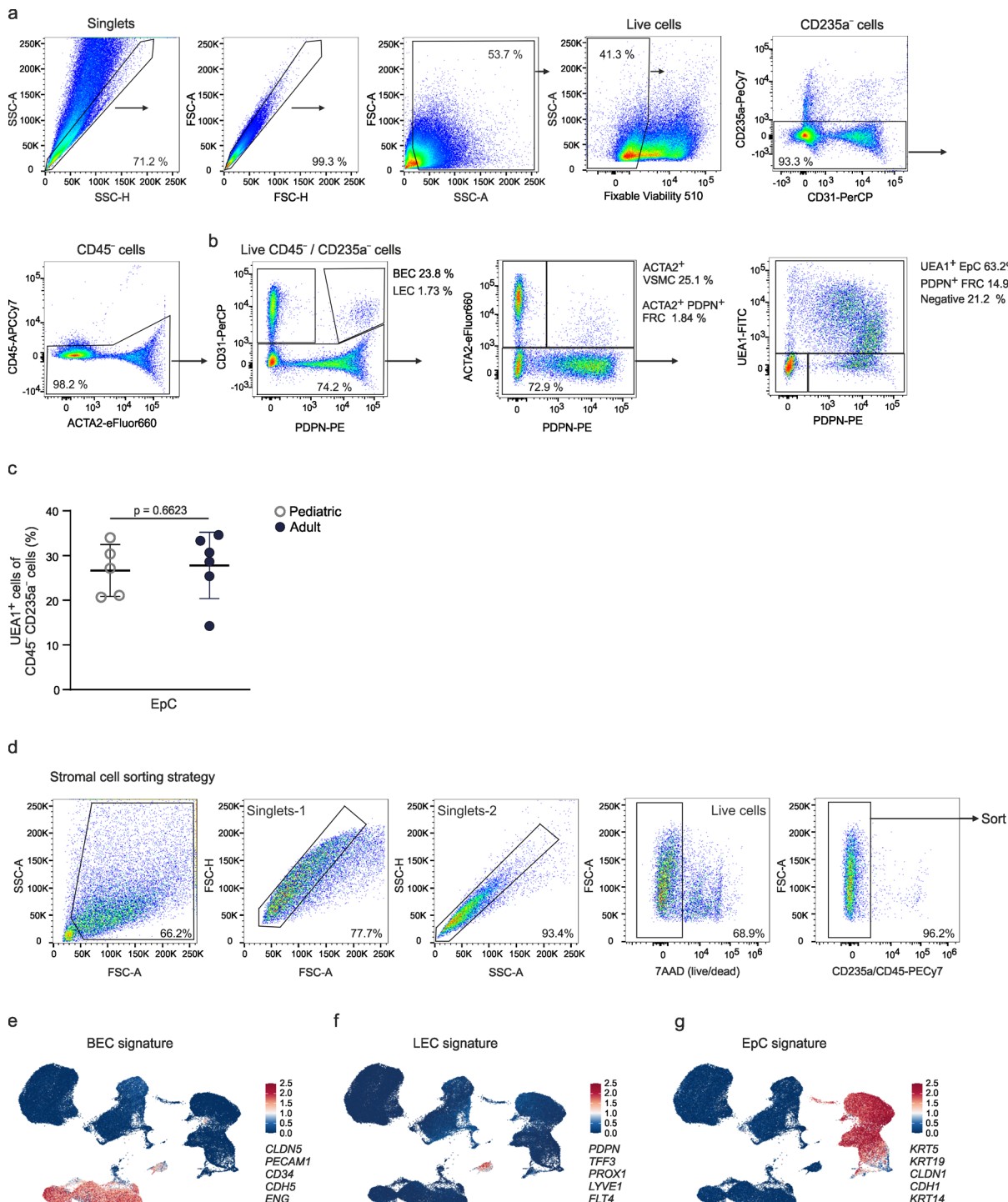

**Extended Data Fig. 3 | Flow cytometric and scRNA-seq analysis of stromal cells in human palatine tonsils. a**,**b**, Representative flow cytometric plots depicting the gating strategy for each stromal cell type according to the indicated markers. **c**, Quantification of epithelial cells (EpC) as a percentage of total CD45⁻ CD235a⁻ live cells in pediatric (*n* = 5) and adult (*n* = 6) patients with OSA. Data show average value of left and right tonsils for each patient. Mean and s.d. are indicated. *P* value was calculated with the two-sided Mann-Whitney test. **d**, Representative flow cytometric plots depicting the sorting strategy for scRNA-seq of CD45⁻ CD235a⁻ stromal cells from human palatine tonsils according to the indicated markers. **e**–**g**, Expression patterns of indicated BEC, LEC and EpC gene signatures projected onto UMAPs. ScRNA-seq data represents a total of 86,966 CD45⁻ CD235a⁻ stromal cells from *n* = 12 patients.

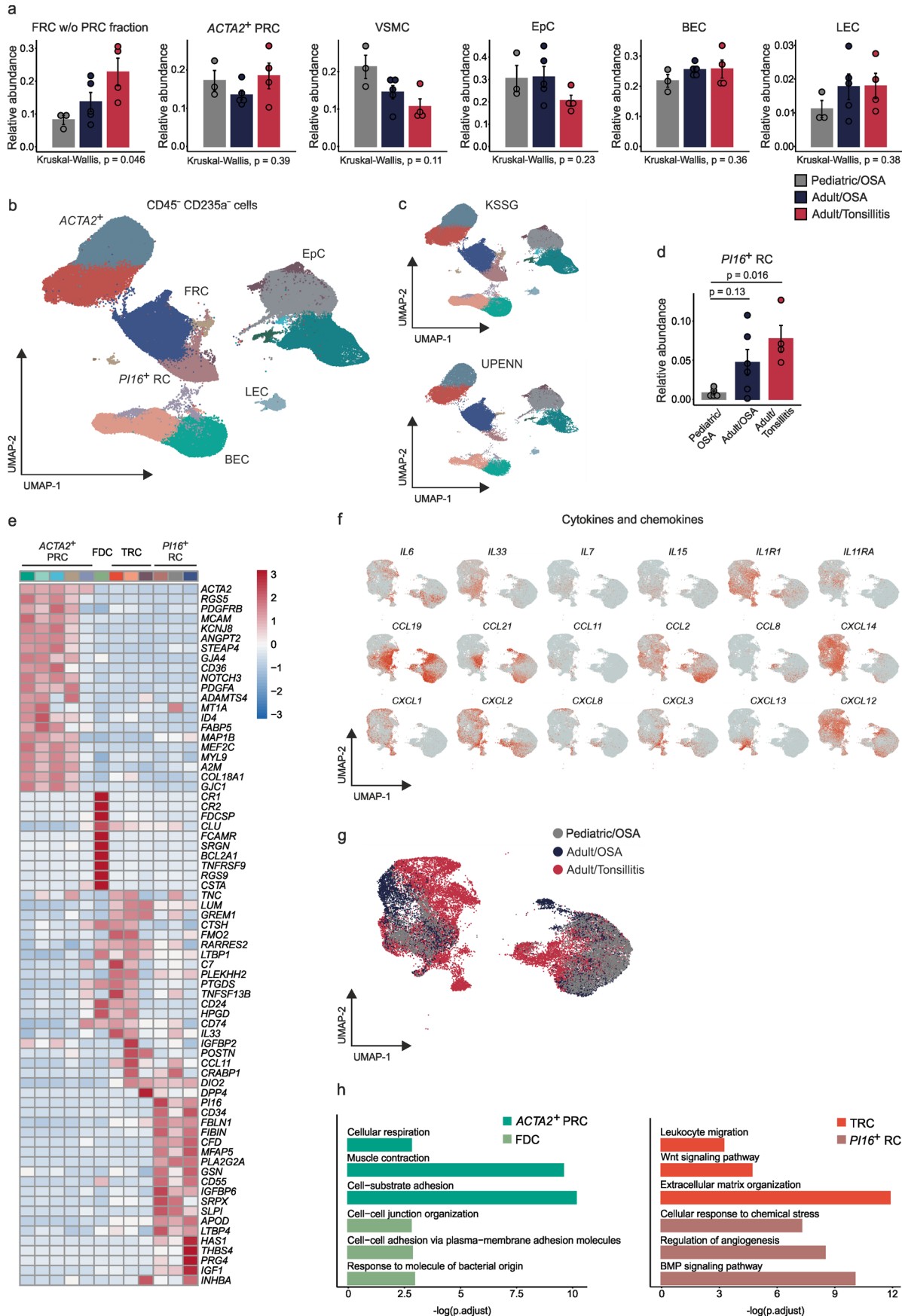

**Extended Data Fig. 4 | See next page for caption.**

**Extended Data Fig. 4 | Age- and inflammation-related molecular changes in tonsillar stromal cells. a**, Bar plots show the relative abundance of indicated stromal cell types among CD45⁻ CD235a⁻ cells of individual patients according to different conditions and based on scRNA-seq data. Mean and SEM are indicated. *P* values as per Kruskal-Wallis test. ScRNA-seq data represents a total of 86,966 CD45⁻ CD235a⁻ stromal cells containing 20,158 cells from *n* = 3 pediatric patients with OSA, 21,596 cells from *n* = 5 adult patients with OSA and 45,212 cells from *n* = 4 adult patients with tonsillitis. **b**, UMAP showing 126,320 CD45⁻ CD235a⁻ stromal cells from 12 patients acquired at KSSG (Kantonsspital St.Gallen, St.Gallen, Switzerland) and 3 patients independently acquired at UPENN (University of Pennsylvania, Philadelphia, USA) (see Extended Data Table 1) integrated over their origin. BECs, LECs, EpCs, *ACTA2*⁺ cells and *PI16*⁺ RCs as a fraction of FRCs are indicated. **c**, UMAP visualization of tonsillar stromal cells split by sample origin. **d**, Bar plot showing the relative abundance of *PI16*⁺ RCs among CD45⁻ CD235a⁻ cells per condition and based on scRNAseq data. Data represents *n* = 5 (Pediatric,OSA), *n* = 6 (Adult,OSA) and *n* = 4 (Adult,Tonsillitis) individual patients. Mean and SEM are indicated. *P* values as per two-sided Wilcoxon test. **e**, Heatmap showing the average expression of marker genes used for characterization of FRC subsets. **f**, Featureplots visualizing the expression pattern of indicated cytokines and chemokines across FRCs. **g**, UMAP depicting re-embedded FRCs colored according to the indicated patient groups. **h**, Significantly enriched terms according to GO enrichment analysis based on differentially expressed genes for the indicated FRC subsets. **e-h**, ScRNA-seq data represents 28,571 FRCs containing 4,867 cells from *n* = 3 pediatric patients with OSA, 6,184 cells from *n* = 5 adult patients with OSA and 17,520 cells from *n* = 4 adult patients with tonsillitis.

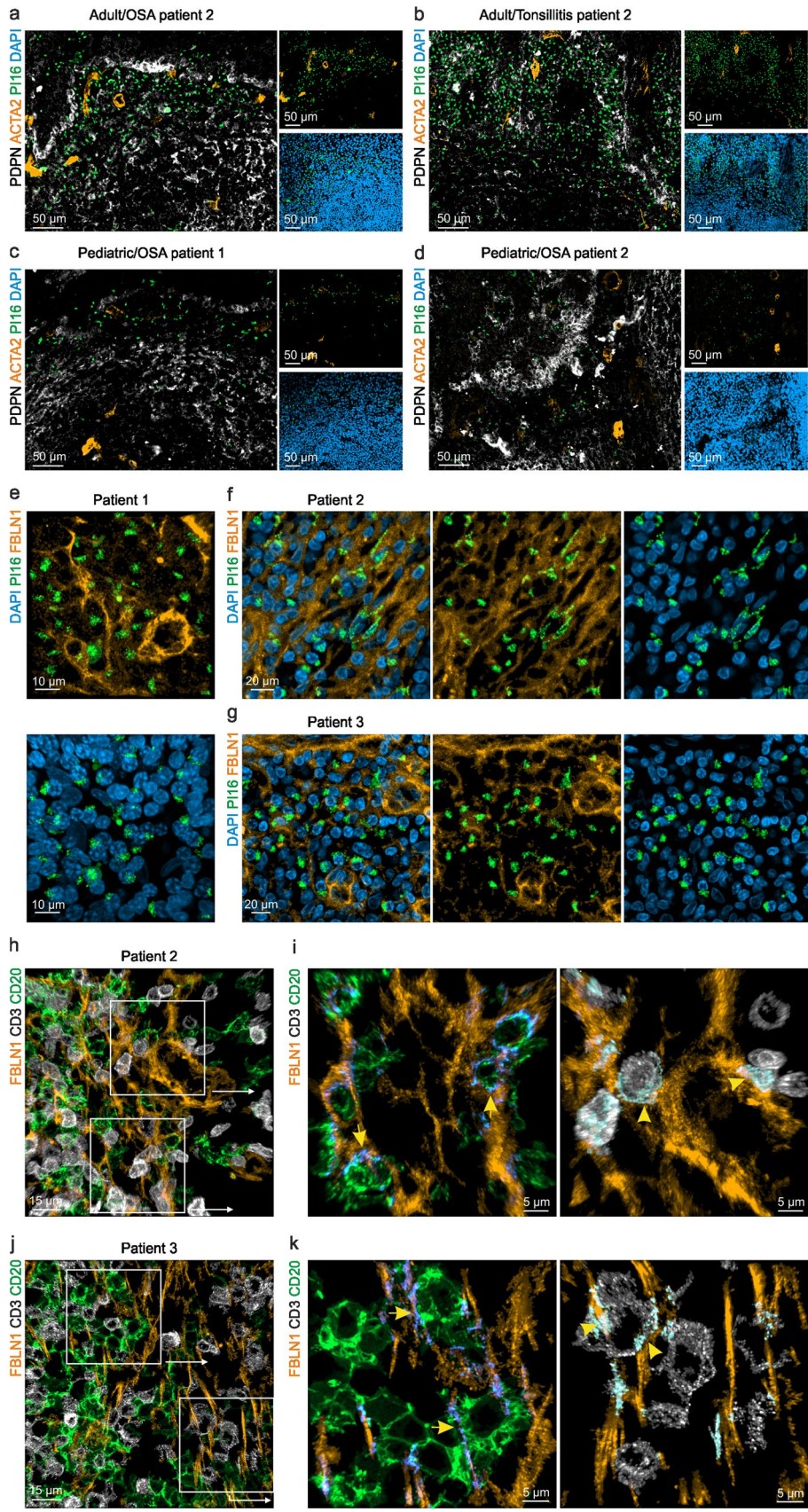

**Extended Data Fig. 5 | See next page for caption.**

**Extended Data Fig. 5 | Histological analysis of subepithelial PI16⁺ RCs. a–d**, PI16-positive cells in the subepithelial niche in tonsil sections from adult patients with OSA (**a**), tonsillitis (**b**) and pediatric patients with OSA (**c**, **d**). Cryosections were stained with indicated antibodies and analyzed by confocal microscopy. Images are representative for $n = 3$ (Pediatric, OSA), $n = 3$ (Adult, OSA) and $n = 3$ (Adult, Tonsillitis) patients. **e–g**, High-resolution microscopy images of adult PI16⁺ cells. Sections were stained with antibodies against FBLN1 and PI16, acquired by confocal microscopy and reconstructed in 3D. **h,j**, Reconstruction of the adult subepithelial FRC niche occupied by CD3⁺ or CD20⁺ lymphocytes. Sections were stained with anti-FBLN, anti-CD3 and anti-CD20 antibodies, acquired by confocal microscopy and reconstructed in 3D. **i,k**, Surface contact areas (blue) of FBLN1⁺ cells with CD20⁺ (arrows) and CD3⁺ (arrowheads) cells are shown at higher magnification. Microscopy images in **e-k** are representative for $n = 3$ adult patients with OSA.

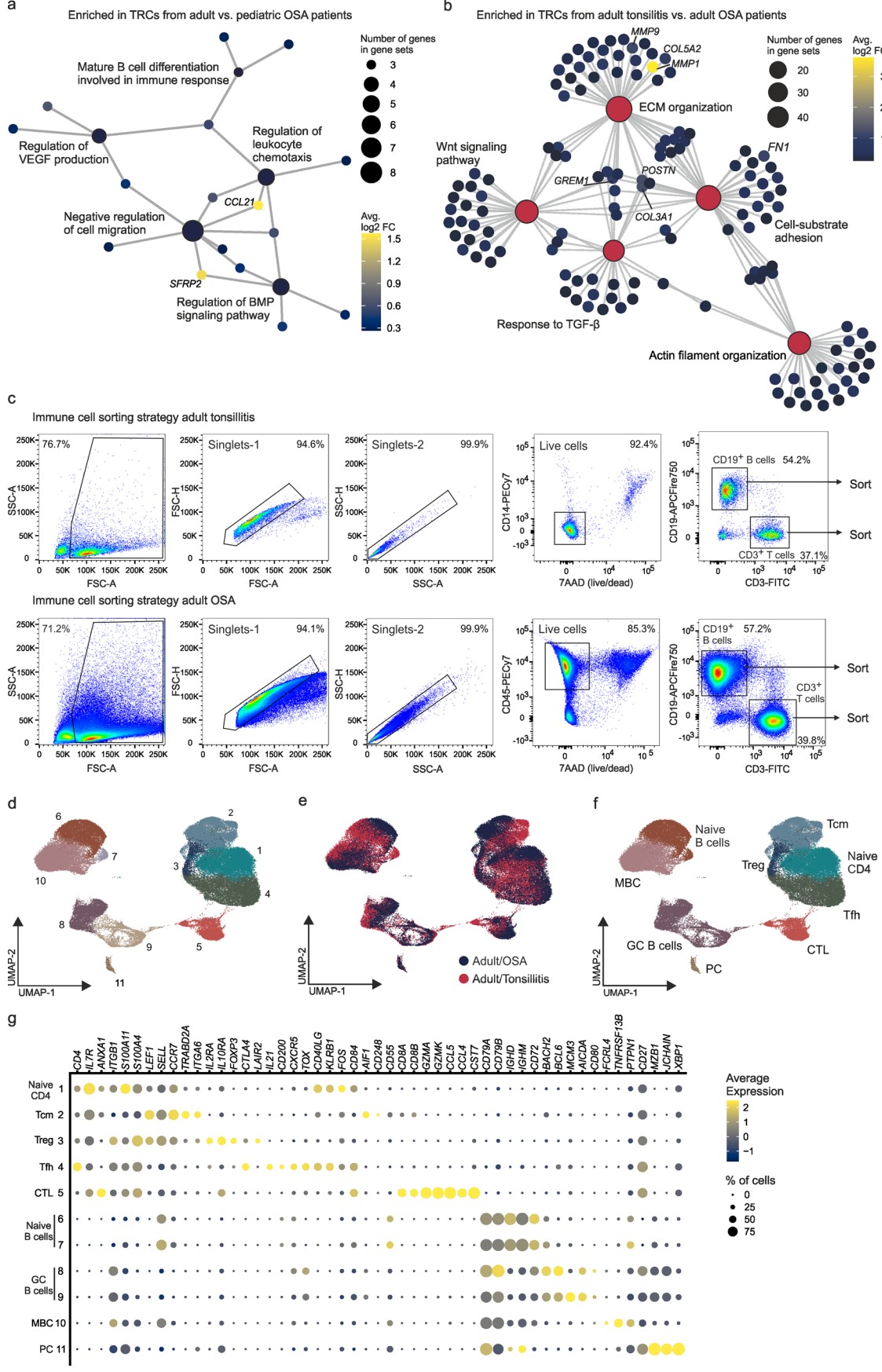

**Extended Data Fig. 6 | See next page for caption.**

**Extended Data Fig. 6 | ScRNA-seq analysis of tonsillar TRCs and immune cells during homeostasis and inflammation. a**, Gene sets enriched in TRCs according to GO enrichment analysis based on differentially expressed genes in adult TRCs compared to pediatric TRCs. **b**, Gene sets enriched in TRCs upon inflammation according to GO enrichment analysis based on differentially expressed genes in TRCs between adult patients with tonsillitis and OSA. Genes with an average log2 fold change (Avg. log2 FC) > 1 assigned to enriched terms are highlighted. **c-g**, ScRNA-seq analysis of the human palatine tonsil T and B cell landscape. **c**, Representative flow cytometric plots depicting the sorting strategy of CD3$^+$ T cells and CD19$^+$ B cells from human palatine tonsils according to the indicated markers. UMAP visualizations show T and B cells from adult patients with OSA and tonsillitis colored according to unbiased clustering (**d**), condition (**e**), and assigned B and T cell subsets (**f**). **g**, Dot plot depicts marker genes used for characterization and assignment of T and B cell subsets. ScRNA-seq data represents a total of 81,997 T and B cells from $n$ = 2 adult patients with OSA and $n$ = 3 adult patients with tonsillitis. Tcm, central memory T cells; Treg, regulatory T cells; Tfh, T follicular helper cells; CTL, cytotoxic T lymphocytes; GC, germinal center; MBC, memory B cells; PC, plasma cells.

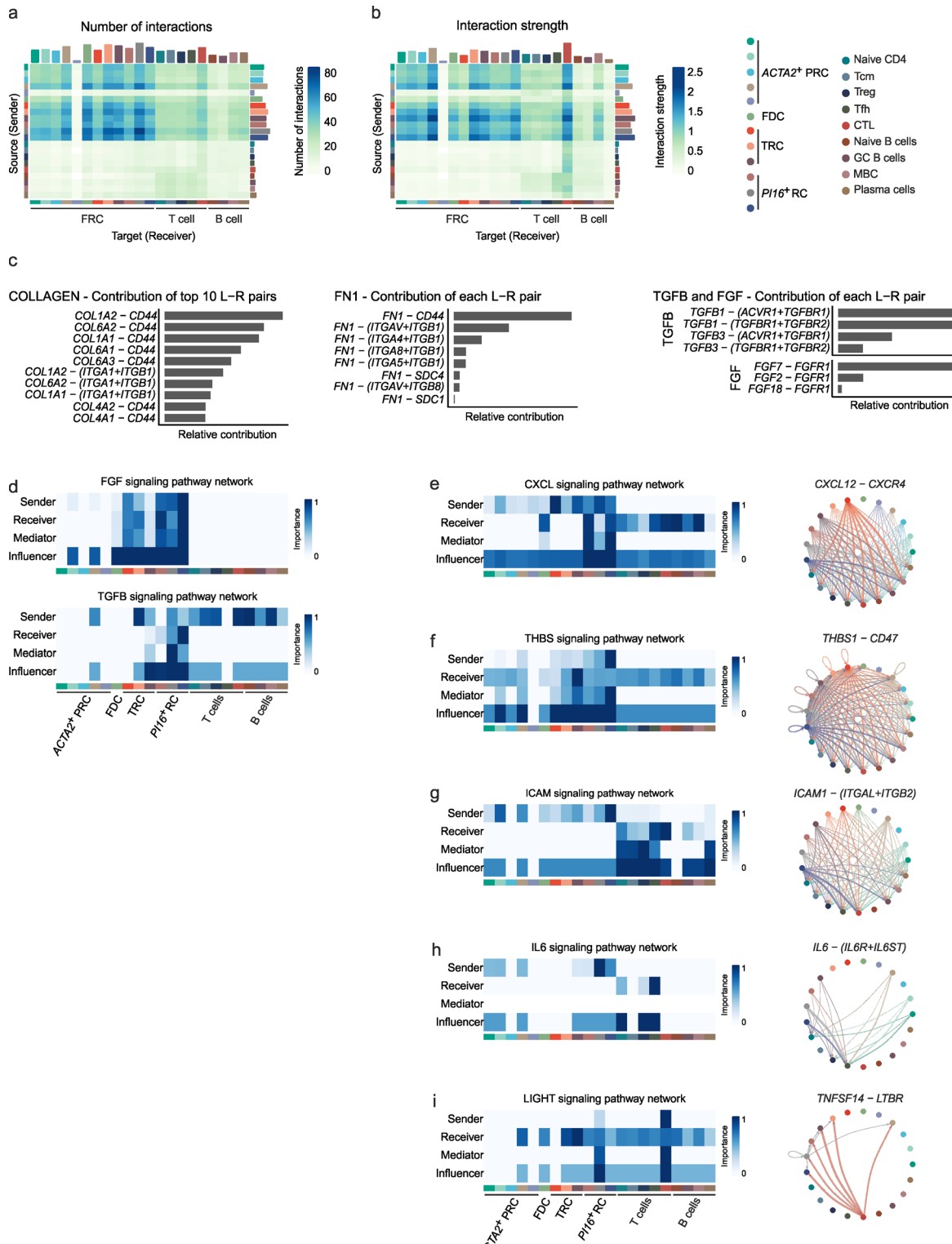

**Extended Data Fig. 7 | Interactome analysis of tonsillar FRCs during homeostasis and inflammation. a,b**, Heatmaps depict outgoing and incoming number of interactions (**a**) and interaction strengths (**b**) of individual cell subsets. Colored bar plots represent the sum of column (incoming signal) or row (outgoing signal) values displayed in the heatmap. **c**, Relative contribution of each ligand-receptor pair to the overall communication network of COLLAGEN, FN1, TGFB and FGF signaling pathways. **d**, Heatmaps show relative importance of each cell subset based on network centrality measures for FGF and TGFB signaling pathway networks. **e-i**, Heatmaps show relative importance of each cell subset based on network centrality measures for CXCL, THBS, ICAM, IL6, and LIGHT signaling pathway networks. Circle plots in the right panel show inferred signaling networks of indicated ligand-receptor pairs. Edge width reflects the communication probability. Interactome analysis represents scRNA-seq data of 81,997 T and B cells from $n$ = 2 adult patients with OSA and $n$ = 3 adult patients with tonsillitis and 23,704 FRCs from $n$ = 5 (Adult,OSA) and $n$ = 4 (Adult,Tonsillitis) patients. Tcm, central memory T cells; Treg, regulatory T cells; Tfh, T follicular helper cells; CTL, cytotoxic T lymphocytes. GC, germinal center; MBC, memory B cells.

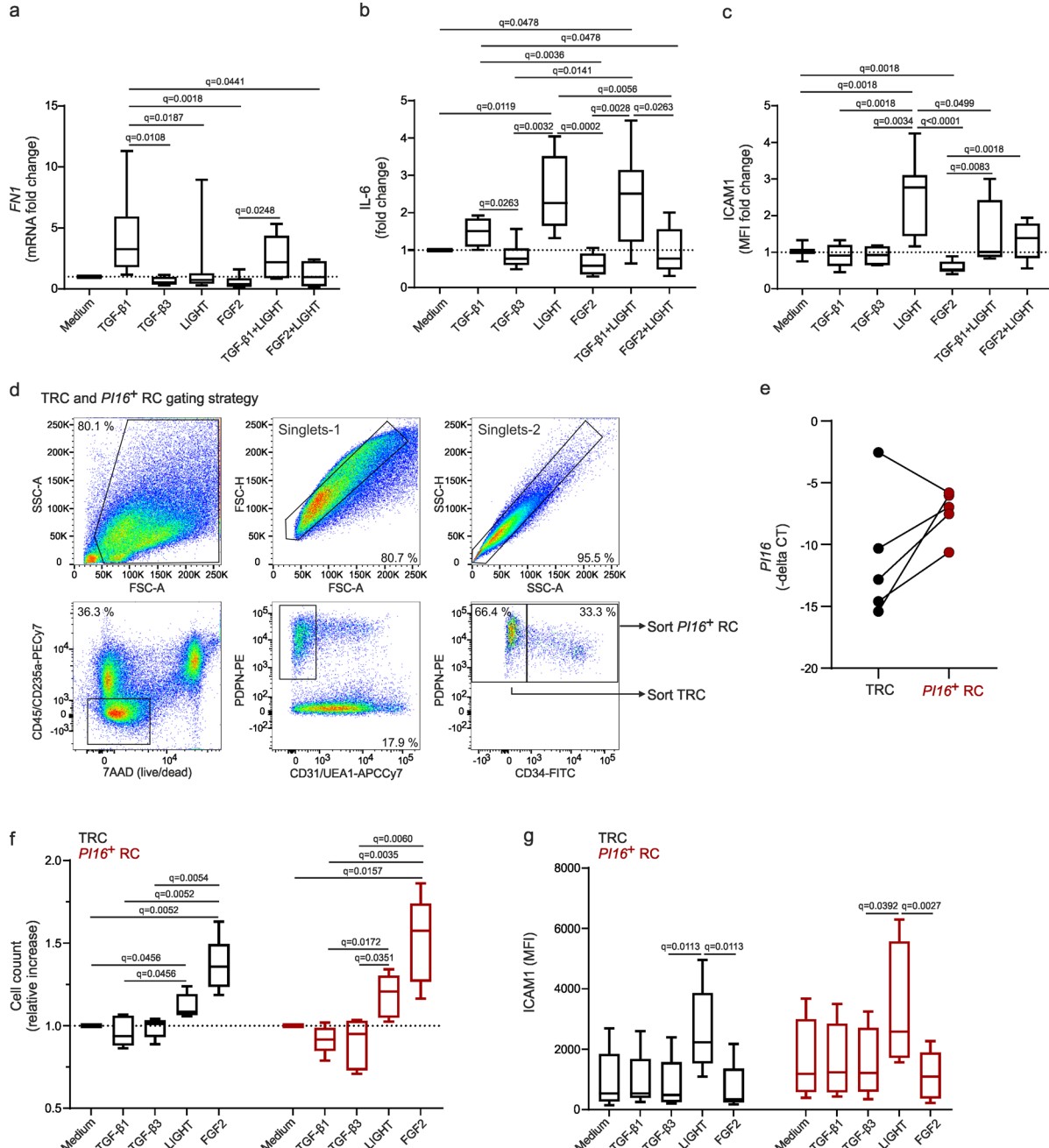

**Extended Data Fig. 8 | In vitro differentiation and activation of tonsillar FRC subsets. a–c**, Bulk cultured primary tonsillar fibroblast of patients with OSA were stimulated with the indicated recombinant proteins for 48 h. **a**, *FN1* mRNA fold change measured by quantitative PCR with reverse transcription (RT-qPCR), n = 8 (Medium, TGF-β1, LIGHT), n = 7 (FGF2), n = 6 (TGF-β1 + LIGHT, FGF2 + LIGHT) and n = 5 (TGF-β3). **b**, IL-6 concentration in culture supernatants. **c**, Flow cytometric analysis of ICAM1 surface expression displayed as fold change of mean fluorescence intensity (MFI). **b**,**c**, n = 8 (Medium, TGF-β1, LIGHT, FGF2) and n = 6 (TGF-β3, TGF-β1 + LIGHT, FGF2 + LIGHT). **d**, Representative flow cytometric plots depicting the sorting strategy for TRCs and *PI16*+ RCs from human palatine tonsils according to the indicated markers, where CD34 was used as a surrogate

marker for *PI16*+ RCs based on the scRNA-seq analysis of tonsillar FRCs. **e**, *PI16* mRNA expression after in vitro expansion of sorted TRCs and *PI16*+ RCs from n = 5 patients with OSA or tonsillitis. **f**,**g**, Sorted TRCs and *PI16*+ RCs from n = 5 adult patients with OSA or tonsillitis were cultured and stimulated with indicated recombinant proteins for 48 h. **f**, Flow cytometric analysis of relative increase in cell counts. **g**, Flow cytometric analysis of ICAM1 mean fluorescence intensity (MFI). **a**-**c**,**f**-**g**, Box extends from 25th to 75th percentile, median is indicated and whiskers show the minimum and maximum values. *Q* values were calculated with the Kruskal-Wallis (**a**-**c**,**f**) or Friedman (**g**) test and Benjamini, Krieger and Yekutieli correction for multiple comparisons.

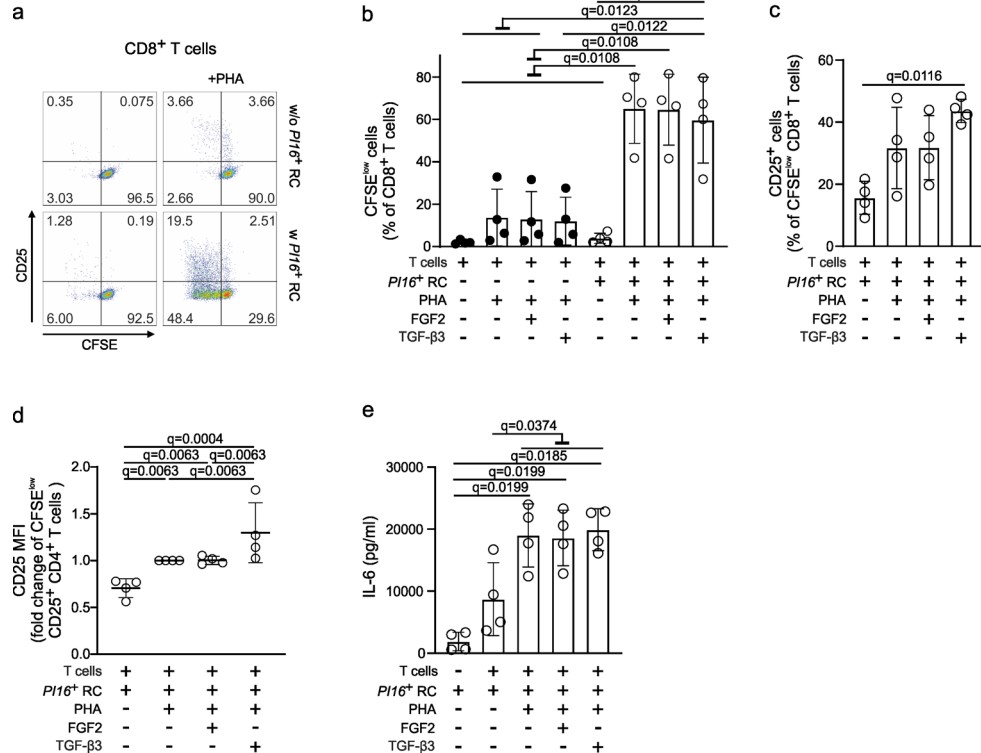

**Extended Data Fig. 9 | Co-culture of *Pl16*⁺ RCs with T cells. a–e,** Carboxyfluorescein succinimidyl ester (CFSE)-labeled tonsillar T cells from adult patients with OSA or tonsillitis were cultured for 3 days with *Pl16*⁺ RCs and indicated recombinant proteins and were stimulated with phytohemagglutinin (PHA). **a**, Representative flow cytometric plots depicting CD25 expression on CFSE^low CD8⁺ T cells. **b**, Quantification of CFSE^low cells as a percentage of CD8⁺ T cells. **c**, Quantification of CD25⁺ cells as a percentage of CFSE^low CD8⁺ T cells. **d**, CD25 mean fluorescence intensity (MFI) fold change of CFSE^low CD25⁺ CD4⁺ T cells. **e**, IL-6 concentration in culture supernatant. **b–e**, Mean and s.d. are indicated, $n = 4$ patients. Q values were calculated with the repeated measure (RM) one-way ANOVA (**b, c, e**) or Ordinary one-way ANOVA (**d**) test and Benjamini, Krieger and Yekutieli correction for multiple comparisons.

**Extended Data Table. 1 | Characteristics of pediatric and adult patients with obstructive sleep apnea or tonsillitis**

| Patient number | Sex[a] | Age[b] (years) | Diagnosis | Origin[c] | scRNAseq data[d] |
|---|---|---|---|---|---|
| Pediatric 01 | F | 13.7 | OSA / Tonsillar hyperplasia | KSSG | Stromal cells[e] |
| Pediatric 02 | M | 5.7 | OSA / Tonsillar Hyperplasia | KSSG | Stromal cells[e] |
| Pediatric 03 | F | 7.8 | OSA / Tonsillar Hyperplasia | KSSG | Stromal cells[e] |
| Pediatric 04 | F | 17 | Chronic tonsillitis | UPENN | Stromal cells[f] |
| Pediatric 05 | F | 6 | Tonsillitis of unclear duration | UPENN | Stromal cells[f] |
| Adult 01 | M | 54.6 | OSA / Tonsillar Hyperplasia | KSSG | Stromal cells[e] |
| Adult 02 | M | 30.4 | OSA / Tonsillar Hyperplasia | KSSG | Stromal cells[e] |
| Adult 03 | M | 37.3 | OSA / Tonsillar Hyperplasia | KSSG | Stromal cells[e] |
| Adult 04 | M | 36.3 | OSA / Tonsillar Hyperplasia | KSSG | Stromal cells[e] / T and B cells |
| Adult 05 | F | 37.9 | OSA / Tonsillar Hyperplasia | KSSG | Stromal cell[e] / T and B cells |
| Adult 06 | M | 21 | OSA / Tonsillar Hyperplasia | UPENN | Stromal cells[f] |
| Tonsillitis 01 | M | 39.1 | Chronic hyperplastic tonsillitis with purulent-ulcerative foci | KSSG | Stromal cells[e] / T and B cells |
| Tonsillitis 02 | M | 19.0 | EBV tonsillitis | KSSG | Stromal cells[e] / T and B cells |
| Tonsillitis 03 | M | 18.3 | Tonsillitis with peritonsillar abscess | KSSG | Stromal cells[e] |
| Tonsillitis 04 | F | 31.8 | Tonsillitis with peritonsillar abscess | KSSG | Stromal cells[e] / T and B cells |

[a] Male to female ratio of 1:4 in pediatric OSA patients; 5:1 in adult OSA patients and 3:1 in tonsillitis patients

[b] Mean age pediatric OSA patients $10.0 \pm 5.1$ years; mean age adult OSA patients $36.3 \pm 11.0$; mean age tonsillitis patients $27.1 \pm 10.2$

[c] KSSG, Kantonsspital St. Gallen, St.Gallen, Switzerland; UPENN, University of Pennsylvania, Philadelphia, USA

[d] T cells, FACS sorted for $CD45^+ CD3^+$ cells; B cells, FACS sorted for $CD45^+ CD19^+$ cells

[e] Stromal cells FACS sorted for $CD45^- CD235a^-$ cells

[f] Stromal cells FACS sorted for $CD45^- EPCAM^-$ cells

# Reporting Summary

## Statistics

For all statistical analyses, confirm that the following items are present in the figure legend, table legend, main text, or Methods section.

| n/a | Confirmed | |
|---|---|---|
| ☐ | ☒ | The exact sample size (*n*) for each experimental group/condition, given as a discrete number and unit of measurement |
| ☐ | ☒ | A statement on whether measurements were taken from distinct samples or whether the same sample was measured repeatedly |
| ☐ | ☒ | The statistical test(s) used AND whether they are one- or two-sided<br>*Only common tests should be described solely by name; describe more complex techniques in the Methods section.* |
| ☐ | ☒ | A description of all covariates tested |
| ☐ | ☒ | A description of any assumptions or corrections, such as tests of normality and adjustment for multiple comparisons |
| ☐ | ☒ | A full description of the statistical parameters including central tendency (e.g. means) or other basic estimates (e.g. regression coefficient) AND variation (e.g. standard deviation) or associated estimates of uncertainty (e.g. confidence intervals) |
| ☐ | ☒ | For null hypothesis testing, the test statistic (e.g. *F*, *t*, *r*) with confidence intervals, effect sizes, degrees of freedom and *P* value noted<br>*Give P values as exact values whenever suitable.* |
| ☒ | ☐ | For Bayesian analysis, information on the choice of priors and Markov chain Monte Carlo settings |
| ☒ | ☐ | For hierarchical and complex designs, identification of the appropriate level for tests and full reporting of outcomes |
| ☒ | ☐ | Estimates of effect sizes (e.g. Cohen's *d*, Pearson's *r*), indicating how they were calculated |

*Our web collection on statistics for biologists contains articles on many of the points above.*

## Software and code

Policy information about availability of computer code

| Data collection | FACSDiva (BD Biosciences, v8.0.1 and v9.0.1), FACSChorus (BD Biosciences, v1.3), ZEN (Zeiss, v3.3) |
|---|---|
| Data analysis | CellChat (v1.1.3), Imaris (v9), Prism (Graphpad v8), ImageJ (Wayne Rasband, 1.49v), FlowJo (Tree Star Inc., v10.6.2), PhenoGraph (v3.0) clustering algorithm, R (v.4.0.0), CellRanger (v3.0.2), R/Bioconductor package scater (v.1.16.0), Seurat R package (v.4.0.1), clusterProfiler R/Bioconductor (v.3.15.3) |

For manuscripts utilizing custom algorithms or software that are central to the research but not yet described in published literature, software must be made available to editors and reviewers. We strongly encourage code deposition in a community repository (e.g. GitHub). See the Nature Portfolio guidelines for submitting code & software for further information.

## Data

Policy information about availability of data

All manuscripts must include a data availability statement. This statement should provide the following information, where applicable:

- Accession codes, unique identifiers, or web links for publicly available datasets
- A description of any restrictions on data availability
- For clinical datasets or third party data, please ensure that the statement adheres to our policy

The scRNA-seq data generated in this study has been deposited in the BioStudies database (www.ebi.ac.uk/biostudies/) and is available under accession code E-MTAB-11715. GRCh38.9 was used as a reference genome to build the indexes. Processed data files can be downloaded from the figshare platform (https://figshare.com) under 10.6084/m9.figshare.21325737 and explored via an interactive browser at https://immbiosg.github.io/FRCdataExplorer/.

# Field-specific reporting

Please select the one below that is the best fit for your research. If you are not sure, read the appropriate sections before making your selection.

☒ Life sciences    ☐ Behavioural & social sciences    ☐ Ecological, evolutionary & environmental sciences

For a reference copy of the document with all sections, see nature.com/documents/nr-reporting-summary-flat.pdf

# Life sciences study design

All studies must disclose on these points even when the disclosure is negative.

| Sample size | No sample-size calculation was performed. Sample sizes were determined to be adequate based on the reproducibility between independent experiments and patients and adequate cell numbers of each subset in the RNA-seq data to run comparative analyses. Sample sizes for both single cell and in vitro experiments were based on our experience and common practise in the field (Nat Immunol. 2020 Jun; 21(6):649-659). |
|---|---|
| Data exclusions | No data points were excluded. |
| Replication | For analysis of the performed scRNA-seq experiments no batch correction needed to be applied for any of the samples. Therefore we can assume careful and good reproducibility. ScRNA-seq data are derived from 12 patients and 9 independent experiments. All attemps at replication were successfull. |
| Randomization | Randomization and control of covariants was not relevant in the setting of this study as there was no intervention performed. |
| Blinding | Blinding was not relevant since data analysis was explorative and no intervention was performed. |

# Reporting for specific materials, systems and methods

We require information from authors about some types of materials, experimental systems and methods used in many studies. Here, indicate whether each material, system or method listed is relevant to your study. If you are not sure if a list item applies to your research, read the appropriate section before selecting a response.

### Materials & experimental systems

| n/a | Involved in the study |
|---|---|
| ☐ ☒ | Antibodies |
| ☒ ☐ | Eukaryotic cell lines |
| ☒ ☐ | Palaeontology and archaeology |
| ☒ ☐ | Animals and other organisms |
| ☐ ☒ | Human research participants |
| ☒ ☐ | Clinical data |
| ☒ ☐ | Dual use research of concern |

### Methods

| n/a | Involved in the study |
|---|---|
| ☒ ☐ | ChIP-seq |
| ☐ ☒ | Flow cytometry |
| ☒ ☐ | MRI-based neuroimaging |

# Antibodies

| Antibodies used | Anti-human PDPN PE (eBioscience, Clone: NZ-1.3, Cat#:  12-9381-42, Lot#: 4332768)<br>Anti-human CD45 PeCy7 (eBioscience, Clone: HI30, Cat#: 25-0459-42, Lot#: 2079970)<br>Anti-human CD31 Biotin (eBioscience, Clone: WM59, Cat#: 13-0319-82, Lot#: 1994108)<br>Anti-human CD31 PerCP (Biolegend, Clone: WM59, Cat#: 303132, Lot#: B272399)<br>Anti-human CD235a PeCy7 (Biolegend, Clone: HI264, Cat#: 349112, Lot#: B274230)<br>Anti-human CD3 FITC (Biolegend, Clone: UCHT1, Cat#: 300440, Lot#: B279209)<br>Anti-human CD14 PE-Cy7 (Biolegend, Clone: MSE2, Cat#: 301813, Lot#: B231081)<br>Anti-human CD19 APC/Fire750 (Biolegend, Clone: HIB19, Cat#: 302258, Lot#:B242981)<br>Anti-human CD45 APC-Cy7 (BD, Clone: 2D1, Cat#: 560178, Lot#: 2079970)<br>Anti-human CD324 APC (Biolegend, Clone: 67A4, Cat#: 324107, Lot#: B263115)<br>Anti-human CD11c A647 (Biolegend, Clone 3.9, Cat#: 301613, Lot#: B208071)<br>Anti-human CD54/ICAM BV421 (Biolegend, Clone: HA58, Cat#: 353132, Lot#: B350347)<br>Anti-human CD34 FITC (Biolegend, Clone: 581, Cat#: 343504, Lot#: B356958)<br>Anti-human CD3 (Dako, polyclonal, Cat#: A0452, Lot#: 20061852)<br>Anti-human PDPN (eBioscience, Clone: NZ-1.3, Cat#: 14-9381-82, Lot#: 2400405)<br>Anti-human CD20 A488 (eBioscience, Clone: L26, Cat#: 53-0202-80, Lot#: 2210882)<br>Anti-human/mouse aSMa eFluor660 (eBioscience, Clone: 1A4, Cat#: 50-9760-82, Lot#: 2060395)<br>Anti-human/mouse aSMa Cy3 (Sigma, Clone:1A4,  Cat#: C6198, Lot#: 0000116745)<br>Anti-human FBLN1 (Atlas Antibodies, Clone: CL0337, Cat#: AMAb906960, Lot#: MAB-03450)<br>Anti-human PI16 (Novus Biologicals, Cat#: NBP1-92254, Lot#: A104691)<br>Anti-human CD4 BUV395 (BD, Clone: SK3, Cat#: 563550, Lot#: 2273355) |
|---|---|

Anti-human CD8 BUV805 (BD, Clone: SK1, Cat#: 612889, Lot#: 1298376)
Anti-human CD25 BUV563 (BD, Clone: 2A3, Cat#: 612918, Lot#: 1028315)

Anti-rabbit IgG A488 (Jackson Immunoresearch, Cat#: 711-545-152, Lot#: 158217)
Anti-rat IgG A488 (Jackson Immunoresearch, Cat#: 712-545-150, Lot#: 150327 )
Anti-mouse IgG A488 (Jackson Immunoresearch, Cat#: 715-545-150, Lot#: 156010)
Anti-rat IgG A594 (Jackson Immunoresearch, Cat#: 712-585-153, Lot#: 152519)
Anti-rat IgG A647 (Jackson Immunoresearch, Cat#: 712-606-150, Lot#: 150018)
Anti-mouse IgG A647 (Jackson Immunoresearch, Cat#: 715-605-150, Lot#: 158693)
Anti-rabbit IgG Cy3 (Jackson Immunoresearch, Cat#: 711-165-152, Lot#: 157936)
Anti-rat IgG Cy3 (Jackson Immunoresearch, Cat#: 712-167-003, Lot#: 144072)
Anti-mouse IgG Cy3 (Jackson Immunoresearch, Cat#: 715-165-150, Lot#: 155993)

| | |
|---|---|
| Validation | All antibodies came from commercial vendors, and were validated by the manufacturers on their official website. For stainings that used a combination of primary and secondary antibodies, each primary antibody was additionally validated by performing control stains using the secondary antibody alone to ensure a specific signal. |

# Human research participants

Policy information about studies involving human research participants

| | |
|---|---|
| Population characteristics | Detailed information is listed in Extended data Table 1 "Characteristics of pediatric and adult patients with obstructive sleep apnea or tonsillitis" |
| Recruitment | Tonsil samples were collected from adult and pediatric patients suffering from obstructive sleep apnea (OSA) or tonsillitis that underwent routine tonsillectomy at the Kantonsspital St. Gallen. Decision to perform surgery was made after clinical assessment by attending ENT physicians. For the tonsils acquired at UPENN (University of Pennsylvania, Philadelphia, USA) and included in the extended cohort tonsil use was approved by a material transfer agreement between the University of Pennsylvania and Children's Hospital of Philadelphia (ID: 58590/00). These tonsils were received without any protected health information or identifiers, exempt from review by the Children's Hospital of Philadelphia Institutional Review Board (SOP 407, Section XII "Secondary Use of De-Identified Data or Specimens"). |
| Ethics oversight | The study protocol has been reviewed and approved by the Ethikkommission Ostschweiz (EKOS), permission numbers 2017-00051 (adult patients) and 2018-01646 (pediatric patients). All study participants provided written informed consent in accordance with the Declaration of Helsinki and the International Conference on Harmonization Guidelines for Good Clinical Practice. All regulations were followed according to the Swiss authorities and according to the clinical protocols. |

Note that full information on the approval of the study protocol must also be provided in the manuscript.

# Flow Cytometry

## Plots

Confirm that:

☒ The axis labels state the marker and fluorochrome used (e.g. CD4-FITC).

☒ The axis scales are clearly visible. Include numbers along axes only for bottom left plot of group (a 'group' is an analysis of identical markers).

☒ All plots are contour plots with outliers or pseudocolor plots.

☒ A numerical value for number of cells or percentage (with statistics) is provided.

## Methodology

| | |
|---|---|
| Sample preparation | A description of the sample preparation for flow cytometry and FACS sorting is detailed in the methods section. |
| Instrument | LSR Fortessa BD Biosciences, FACS Melody BD Biosciences, FACSymphony A3 BD Biosciences |
| Software | FACSDiva (BD Biosciences, v8.0.1 and v9.0.1) was used to collect the data and FlowJo software v10.6.2 (Tree Star Inc.) to analyze the data. FACSChorus (BD Biosciences, v1.3) was used to set up cell sorting, and R v.4.0.0 was used to analyze the transcriptomic data. |
| Cell population abundance | High purity of the post-sort fraction was confirmed by downstream scRNA-seq analysis. |
| Gating strategy | For all flow cytometric analysis, cells were first gated on FSC/SSC to exclude cell debris following by FSC-A/FSC-H and SSC-A/SSC-H to exclude doublets. Dead cells were excluded from analysis by gating on viability dye negative staining. Gating strategy for identifying stromal cell populations in this study is exemplifying in the Extended data figures. |

☒ Tick this box to confirm that a figure exemplifying the gating strategy is provided in the Supplementary Information.

