## [Peer Review File · Nature Immunology]

Peer Review Information

Journal: Nature Immunology

Manuscript Title: PI16+ reticular cells in human palatine tonsils govern T cell activity in distinct subepithelial niches

Corresponding author name(s): Professor Burkhard Ludewig

Reviewer Comments & Decisions:

Decision Letter, initial version:
--

9th Jun 2022

Dear Burkhard,

Good to see you at the Aegean conference and to discuss with you & your students the proposed revisions to your study entitled "PI16+ reticular cells in human palatine tonsils activate T and B cells in distinct subepithelial niches". As noted by the comments from the referees, while we cannot accept the current version, we would very much be interested in a revised manuscript that addressed the referees' concerns.

We invite you to submit a substantially revised manuscript, however please bear in mind that we will be reluctant to approach the referees again in the absence of major revisions.

Specifically, the revision should include new experiments to address:

(1) look at additional human tonsil sampled to sort Pi16+ reticular cells and TRCs and interrogate functional roles by in vitro culture/stimulation

(2) obtain additional acutely inflamed tonsils for additional scRNA-seq analysis

(3) provide additional representative images of tonsils

Also provide URL links for user-friendly web-based platforms for dataset exploration.

Please include the additional textual clarifications as indicated in your response letter.

When you revise your manuscript, please take into account all reviewer and editor comments, please highlight all changes in the manuscript text file in Microsoft Word format.

* If you have not done so already please begin to revise your manuscript so that it conforms to our Article format instructions at <http://www.nature.com/ni/authors/index.html>. Refer also to any guidelines provided in this letter.

The Reporting Summary can be found here:

When submitting the revised version of your manuscript, please pay close attention to our [href="https://www.nature.com/nature-portfolio/editorial-policies/image-integrity">Digital Image Integrity Guidelines. and to the following points below:](https://www.nature.com/nature-portfolio/editorial-policies/image-integrity)

[REDACTED]

If you wish to submit a suitably revised manuscript we would hope to receive it within 6 months. If you cannot send it within this time, please let us know. We will be happy to consider your revision so long as nothing similar has been accepted for publication at Nature Immunology or published elsewhere.

Nature Immunology is committed to improving transparency in authorship. As part of our efforts in this direction, we are now requesting that all authors identified as 'corresponding author' on published papers create and link their Open Researcher and Contributor Identifier (ORCID) with their account on

the Manuscript Tracking System (MTS), prior to acceptance. ORCID helps the scientific community achieve unambiguous attribution of all scholarly contributions. You can create and link your ORCID from the home page of the MTS by clicking on 'Modify my Springer Nature account'. For more information please visit www.springernature.com/orcid.

Thank you for the opportunity to review your work.

Kind regards,

Laurie

Laurie A. Dempsey, Ph.D.
Senior Editor
Nature Immunology
l.dempsey@us.nature.com
ORCID: 0000-0002-3304-796X

Referee expertise:

Referee #1: Human immunology

Referee #2: B cells

Referee #3: Germinal center reactions

Reviewers' Comments:

Reviewer #1:

Remarks to the Author:

In the current manuscript, De Martin, Stanossek, and co-workers provide a detailed histological and transcriptomic overview of palatine tonsils. Tissues were isolated from pediatric and adult OSA patients and supplemented with a single case of adult tonsillitis.

The manuscript is extremely well written, the imaging beautiful, and the scRNA seq analysis well done, but the manuscript falls short in the generation of mechanistic insight. The manuscript does not provide a single experimental confirmation of the predicted functions of the PI16+ reticular cells. Most conclusions, including the title, abstract, conclusion sentences in results section and discussion are worded too strongly, as all conclusions are speculative and only based on in-silico predictions. While I realize that mechanistic research into immune processes in human are not trivial, its absence significantly reduces the impact of the study.

The results and conclusions on the effect of inflammation, which comprise a fair portion of the second

half of the manuscript, are based on a single case of tonsillitis. It is impossible to know whether any of the observed effects are inflammation-specific or patient-specific and as such should their merit is impossible to judge.

Specific points

Throughout the manuscript it is unclear how reproducible the observed differences between adult and pediatric tonsils are. What is the variation between donors?

The manuscript relies heavily on bioinformatics analyses. Unfortunately, on several occasions, the text describing the results suggests differences that for non-bioinformatics experts are difficult to reconcile with the figures. For instance: "The transcriptional profile within FRC populations changed profoundly in adult tonsils (Fig. 4b and c) compared to the pediatric tonsils (Fig. 4a)." (lines 227-229). These profound changes are not easily detected in the UMAP representations and it would help the reader if the authors could explain in more detail the differences they are referring to.

Due to the current selection of images, conclusions on interactions between FBLN1 stromal cells and T and B cells appear to be based on 2 images re-used in multiple figures. Figure 5C is re-used extended fig 5b, the same goes for figures 5e and extended figure 5c. The panels in figure 5f are enlargements of the panels in extended figure 5d. The use of different micrographs in different figures would help to provide a sense of reproducibility of the co-localization.

Reviewer #2:

Remarks to the Author:

In this study De Martin et al., perform a detailed characterization of the stromal populations in human palatine tonsils. Although human tonsils have been used in research for many years, the stromal cell populations and their organization has not been comprehensively studied. This manuscript includes well performed immunofluorescence microscopy revealing some new marker features of the squamous epithelium overlying the tonsil, including showing a basal layer of crypt epithelium expresses PDPN. A four marker FACS panel is developed to permit quantitation of the major stromal cell types. Comparison of pediatric and adult tonsil reveals a reduction in the PDPN+ACTA2+ subset with aging. scRNAseq of CD45-CD235a- tonsil cells is used to interrogate the gene expression profile of the different stromal types. Interestingly, the ACTA2+ population that contains perivascular (smooth muscle) cells also contains cells with FRC properties (termed perivascular reticular cells or PRC). Assessment of cells from one adult tonsillitis patient (compared to cells pooled from 5 adult OSA patients and 3 pediatric OSA patients) revealed an increase in FRC and PRC content. PI16-expressing RCs were found to make up a fraction of the FRCs (confirmed by IF) and to be enriched in adult tonsil and undergo changes in gene expression during tonsillitis. GSEA indicated these cells may have functions related to angiogenesis, cell stress response and BMP signaling. Finally, scRNAseq is performed on lymphocytes from two adult OSA patients and the tonsillitis patient. All the data were then used in CellChat analysis to examine the likely types of communications of stromal cells with each other and with lymphocytes. These data provided support for some communications that have been genetically and functionally defined in the mouse, and they suggest some unstudied communications such as maintenance of PI16+ RC by lymphocyte TGFb-derived signals.

Overall, this work adds a significant amount of new descriptive data and some new insight beyond the multitude of past studies on tonsil anatomy and lymphoid composition. The evidence that adventitial

PI16+ RC may not be precursor cells is notable and likely of general interest as it contrasts with the suggestion made in Buechler et al., 2021 (ref 37). A general concern I have with the study is that no attempt is made to use in vitro (or mouse model) experiments to confirm any of the suggested functional relationships identified by the scRNAseq approaches.

Specific concerns

1. The title and abstract imply some previously undefined subepithelial compartment has been identified. However, the compartments described appear in line with those detailed in prior studies. What is newly defined is greater heterogeneity in the FRCs than indicated by prior work, in particular PI16+ RCs and PRCs, within these lymphoid regions.
2. Can we be confident that data obtained from a single tonsillitis patient are representative of common changes during inflammation?
3. The scRNAseq data, as with past data studies by this group, has been deposited as primary data as permitted by the ArrayExpress database. However, the more widely used GEO database requires processed data to be deposited. It can be challenging for labs that lack advanced computing power to download the raw data and run the 10x cellranger pipeline themselves, especially if they are just wanting to quickly query the data. It is strongly recommended that processed data are deposited. Moreover, many studies that involve largely scRNAseq data acquisition and analysis include a user-friendly portal to facilitate data querying. Such a portal is likely to increase the utility and citation of the study.
4. Gene expression data are used extensively to make statements indicating a functional activity has been established. For example, on line 257 'PI16+ RCs exhibited an array of functions' and on lines 469-471 'we found PI6+ RCs utilize canonical immune cell-stimulating pathways...to increase immune cell activation'. While it may be reasonable to use mRNA expression data to suggest that cells may have certain functions, they are not sufficient to conclude that they have those functions.
5. No attempt is made in the results or discussion to put the findings in the context of the Mourcin et al., study (ref 29) that used markers such as CD49a to facilitate tonsil FRC subsetting. This lack of sufficient credit for prior work is even more notable when it is considered that Mourcin et al., included some cell culture experiments to test functional properties of tonsil FRC subsets such as their ability to respond to TGFb.
6. This study reports a lack of marginal reticular cells (MRC) based on scRNAseq. A lack of MRC in tonsil was previously reported by Mourcin et al., based on a lack of TRANCE staining (the conventional approach to identify MRC; TRANCE staining in human LN was confirmed).
7. The findings should also be related to the work of Heesters et al., JEM 2021 characterizing FDC from human tonsil. Based on a brief Pubmed search, the lack of literature citation extends to other studies. For example, FGFs are suggested to contribute to FRC maintenance in an autocrine manner. Yet a study showing that tonsil stromal cell proliferation is promoted by FGF5 is not cited (Park et al., 2016 PMID 27224250). PMID 27907202 also contains relevant information on FACS analysis of tonsil stromal cells and their isolation and response to inflammatory stimuli.
8. It is commented "Our single cell transcriptomics analysis did not provide evidence that specialized, microfold (M) cell-like epithelial cells, as suggested in previous studies^{17, 18}, are part of the lymphoreticular epithelium." This sentence is poorly worded. The data in ref 17 shows convincing evidence of M cells in human tonsil epithelium. This discrepancy needs more discussion. PMID 20141577 may also be relevant.

Minor

1. UEA1 is introduced as an endothelial marker but the initial images show the epithelial expression. The text could be changed for clarity.

2. Line 102, inlet should be inset.

3. Figure 5e and Extended data Figure 5c are the same image. Also, Extended data Figure 5d seems redundant with what is shown in Figure 5e and f.

Reviewer #3:

Remarks to the Author:

Ludewig and colleagues use imaging, flow cytometry and scRNAseq to characterise the human tonsil in a high level of detail, with a specific focus on the non-hematopoietic cells that are essential for orchestrating immune responses in secondary lymphoid tissues. The strength of this paper is the focus on stromal cells in human secondary lymphoid organs, which are not given the appropriate amount of attention, and the depth of the phenotyping of these cells. The weaknesses of this paper are that it is descriptive and most of the interesting insights from scRNAseq are not followed up.

Major points to consider:

The images presented in figure 1 are very beautiful and provide a nice description of the tonsil. The challenge with this type of data (where one only representative image is shown) is to present data from multiple individuals to convey how typical these features are. One approach is to quantify key features from the imaging in multiple people and show these data, or to include additional images from multiple people in the extended figures.

The use of the word ageing is problematic. Its most common use pertains to older age or "elderly". Here, there is a nice comparison of paediatric and adult tonsil samples, but none of the adults can be over 53 years of age. Therefore, the age-based comparison is more relevant to childhood vs. adulthood, rather than the ageing process per se. I would recommend that the use of the word ageing is reconsidered so that the key messages are clear.

The statistical analyses in Figure 2c-e and Extended data 2 are not appropriate. Each symbol is a single tonsil, with two coming from each person. The Mann-Whitney assumes that all datapoints are independent, which they are not as two points come from the same person. It would be more appropriate to have one point per person (possibly the average of the two tonsils). Also, for this data type the standard deviation is more appropriate to show than the SEM, as the latter is relevant for showing the spread of different populations, rather than different individuals.

Figures 3, 4 and 6 utilise single cell RNAseq to further characterise the stromal cell populations in the tonsils. Further defining changes between children and adults, and using one sample, someone with tonsillitis. With the exception of staining PI16+ cells in the tonsil by confocal there is no validation or further exploration of these data. Potential interacting partners are identified bioinformatically, which is interesting, but with no biological follow up this is more hypothesis generating than providing new knowledge about how tonsillar cells interact with each other. Some in vitro co-culture studies, or following up the TGFb observations with some phospho-staining (e.g. figure 3 of this paper: <https://www.ncbi.nlm.nih.gov/pmc/articles/PMC4183221/>) would really raise the interest of these observations.

Minor points to consider:

Please update Extended table 1 to have each patient as a row, and include the age (in years), sex and reason for tonsillectomy for each person.

Extended figure 2. There appears to be a compensation issue with CD45-APC-Cy7 and ACTA2 APC, can this be resolved in flowjo to improve data quality. This will be important for any automated grouping analyses like the UMAP analysis in figure 2. The gate on the live cells is also very generous (fixable viability 510) is there a reason for this? Could the data be improved with a stricter gating here.

Author Rebuttal to Initial comments

See inserted PDF

Dear Laurie, Dear Editors of Nature Immunology,

We would like to thank the reviewers and editors of Nature Immunology for the opportunity to revise our manuscript entitled “PI16+ reticular cells in human palatine tonsils activate T and B cells in distinct sub-epithelial niches”. We highly appreciate the positive feedback and constructive suggestions of the reviewers. We have productively addressed all of the reviewers’ comments and have provided our responses in the point-by-point reply including references to the corresponding changes in figures and the main text.

Overall, the revised manuscript contains amended items (e.g. increased numbers of patients/samples) and new items with data from specifically designed validation experiments:

- Figures:
 - o Fig. 2 containing 5 amended items (a-e)
 - o Fig. 3 containing 10 amended items (a-j)
 - o Fig. 4 containing 9 amended items (a-i)
 - o Fig. 6 containing 11 new/amended items (a-k)
 - o Total: 4 revised Figures containing 35 new/amended items
- Extended Data Figures:
 - o Extended Data Fig. 1 containing 2 amended items (d,e)
 - o Extended Data Fig. 2 containing 8 amended items (a-h)
 - o New Extended data Fig. 3 containing 3 amended items (a-c)
 - o New Extended data Fig. 4 containing 3 amended items (b-d)
 - o New Extended data Fig. 5 containing 8 amended items (a-h)
 - o New Extended data Fig. 6 containing 9 amended items (a-b,d, f-k)
 - o New Extended data Fig. 7 containing 6 amended items (a-b, d-g)
 - o New Extended data Fig. 8 containing 7 amended items (a-g)
 - o New Extended data Fig. 9 containing 9 new items (a-i)
 - o Total 9 new Extended Data Figures containing 55 amended/new items
- Extended Data Table:
 - o New Extended Data Table 1

Reviewer 1:

In the current manuscript, De Martin, Stanossek, and co-workers provide a detailed histological and transcriptomic overview of palatine tonsils. Tissues were isolated from pediatric and adult OSA patients and supplemented with a single case of adult tonsillitis.

The manuscript is extremely well written, the imaging beautiful, and the scRNA seq analysis well done, but the manuscript falls short in the generation of mechanistic insight. The manuscript does not provide a single experimental confirmation of the predicted functions of the PI16+ reticular cells. Most conclusions, including the title, abstract, conclusion sentences in results section and discussion are worded too strongly, as all conclusions are speculative and only based on in-silico predictions.

- 1) *While I realize that mechanistic research into immune processes in human are not trivial, its absence significantly reduces the impact of the study.*

We thank Reviewer 1 for the productive comments and agree that validation of the predicted interactions and mechanisms is important to strengthen the conclusions of our study. To address this point experimentally, we have used an in vitro differentiation approach to delineate the molecular cues that shape the PI16⁺ RC niche.

To establish whether the predicted ligands/stimulatory factors affect tonsillar FRCs, we first expanded bulk tonsillar fibroblasts and stimulated the cells in vitro with recombinant human TGFβ1, TGFβ3, LIGHT

(TNFSF14) and FGF2. We found that TGFβ1 strongly affects PI16⁺ RC differentiation as revealed by qRT-PCR assessment of *PI16* expression in tonsillar fibroblasts from eight OSA patients (new Fig. 6k). Likewise, TGFβ1 induced the expression of *FN1* (encoding for fibronectin), a factor that was predicted to be highly expressed in PI16⁺ RCs and to be upregulated during inflammation (new Extended Data Fig. 9a). TGFβ1 and LIGHT fostered IL-6 secretion by tonsillar fibroblasts, whereas FGF2 down-regulated IL-6 production (new Extended Data Fig. 9b). In addition, LIGHT induced upregulation of ICAM surface expression, whereas FGF2 reduced ICAM expression (new Extended Data Fig.9c). Stimulation with both TGFβ1 and LIGHT showed that these factors balance the inflammatory reaction of tonsillar fibroblasts as exemplified by the down-regulation of ICAM surface expression in comparison to LIGHT stimulation alone (new Extended Data Fig. 9c). Noteworthy, FGF2 appears to antagonize the effects of TGFβ1 on *PI16/FN1* expression and of LIGHT on IL-6 production and ICAM expression (new Fig. 6k and new Extended Data Fig. 9a-c). Overall, these data confirm that TGFβ, FGF and LIGHT are important factors that determine the phenotype and function of tonsillar fibroblasts.

To further assess to what extent TGFβ1, TGFβ3, LIGHT and FGF2 specifically shape the subepithelial niche generated by PI16⁺ RCs, we have sorted PI16⁺ RCs and TRCs as a control population from adult OSA and tonsillitis patients (N=5). The approach using CD34 and PDPN staining in combination with CD31, UEA1 and CD45 labeling of endothelial, epithelial and hematopoietic cells, respectively, is depicted in the new Extended Data Fig. 9d. Sorted cells were expanded in culture for 10-20 days and their cell identity and morphology was assessed using *PI16* qRT-PCR (new Extended Data Fig. 9e) and confocal microscopy (Fig. R1 for the attention of the reviewer). The in vitro cultured FRC subsets were stimulated with TGFβ1, TGFβ3, LIGHT (TNFSF14) and FGF2 proteins for 48 h. We found that FGF2 serves as a proliferation factor for TRCs and PI16⁺ RCs (new Extended Data Fig. 9f). LIGHT stimulation yielded activation of both TRCs and PI16⁺ RCs leading to increased ICAM surface expression and IL-6 production (new Extended Data Fig. 9g and h). In contrast, TGFβ3-dependent stimulatory effects appeared to be restricted to PI16⁺ RCs leading to increased IL-6 production (new Extended Data Fig. 9h). Moreover, both TGFβ1 and TGFβ3 induced *THBS1* expression mainly in PI16⁺ RCs as predicted by the bioinformatics analysis (new Extended Data Fig. 9i).

In sum, the in vitro analyses using bulk preparations of tonsillar fibroblasts and sorted PI16⁺ RCs and TRCs underscore the conclusion that TGFβ signaling serves as one of the key pathways that control the establishment of the PI16⁺ RC niche. Moreover, the data confirm the predicted activities of FGF and TNFSF14/LIGHT on expansion and activation of PI16⁺ RCs, respectively.

Changes in the main and extended data figures have been indicated in the above text. Methods section and figure legends of the manuscript have been amended accordingly.

Fig. R1. Histological analysis of cultured PI16⁺ RCs and TRCs. *a-b*, Morphology of in vitro cultured PI16⁺ RCs (*a*) and TRCs (*b*) as determined by confocal microscopy using the indicated markers with identical staining and acquisition settings.

- 2) *The results and conclusions on the effect of inflammation, which comprise a fair portion of the second half of the manuscript, are based on a single case of tonsillitis. It is impossible to know whether any of the observed effects are inflammation-specific or patient-specific and as such should their merit is impossible to judge.*

We have addressed this valid concern and have been able to acquire acutely inflamed tonsillar tissue from two patients undergoing partial tonsillectomy due to bacterial infection. In addition, we have received tonsillar tissue from one patient who underwent emergency tonsillectomy due to EBV-associated tonsillitis. The patient characteristics of all patients included for single cell transcriptomics are listed in the new Extended Data Table 1. The scRNA-seq analysis with the additional datasets of three tonsillitis patients comprises now 86,966 CD45⁻ CD235a⁻ stromal cells from 3 pediatric OSA, 5 adult OSA and 4 adult tonsillitis patients. The data and display items in Figures 3, 4 and 6 have been amended accordingly. Unsupervised clustering and UMAP visualization of the extended dataset revealed 14 stromal cell clusters (Fig. 3b). Consistent with the previous analysis, computation of cluster-specific genes revealed transcriptional signatures of BECs (*PECAM1*, gene coding for CD31); LECs (*PECAM1* and *PDPN*); FRCs based on the expression of *PDPN*, *COL3A1* and the chemokines *CCL19* and *CXCL13*; smooth muscle cells (SMCs) expressing *MYF5* and *DES*; *ACTA2*-expressing cells; and EpCs sharing expression of *CDH1* (gene encoding for CD324), *CLDN1*, *KRT5* and *KRT19* (Fig. 3b-c and Extended Data Fig. 3b-d). In addition, we have detected an additional cluster of skeletal muscle cells (SkMC) expressing *TNNT1* and *TNNC1*, which are most probably derived from muscle tissue in proximity to palatine tonsils excised during the challenging surgical procedures. The analysis of the extended datasets confirmed previously established VSMC (Fig. 3d) and FRC gene signatures (new Fig. 3e) and revealed *ACTA2*⁺ cells in cluster 8 as perivascular reticular cells (PRCs) and *ACTA2*⁺ cells in cluster 9 as VSMCs. Gene set enrichment analysis based on genes shared between *ACTA2*⁺ PRCs and VSMCs further confirmed *ACTA2*⁺ PRC and VSMC identity (Figure 3g-j). Overall, the analysis of the extended datasets based on the inclusion of additional tonsillitis patients confirms the cellular identity of tonsillar PRCs.

Moreover, the new analyses corroborated the profound transcriptional changes in FRCs in inflamed tonsils and the associated relative increase in FRCs (Fig. 4a-d and new Extended Data Fig. 5a). Re-analysis of the age- and inflammation-associated changes in the FRC transcriptome revealed twelve different clusters at the pre-defined resolution (Fig. 4e). Cluster-specific gene expression analysis substantiated grouping into four major FRC subsets: *ACTA2*⁺ PRCs, follicular dendritic cells (FDCs), T cell zone reticular cells (TRCs) and *PI16*⁺ RCs (Extended data Fig. 5e). Analysis of the extended datasets confirmed that the presence of *PI16*⁺ RCs was almost completely restricted to adult tonsils and that inflammation substantially affected the transcriptional phenotype of this particular FRC subset (Fig. 4g and Extended data Fig. 5g).

To further substantiate the almost complete absence of *PI16*⁺ RCs in tonsils of pediatric patients, we incorporated three independent scRNA-seq datasets obtained from collaborators at the University of Pennsylvania (UPenn) (two pediatric patients who underwent tonsillectomy due to chronic tonsillitis and one adult patient with obstructive sleep apnea (new Extended data Table 1)). ScRNA-seq datasets from UPenn have not been included into the main analyses because lower sequencing depth led to batch effects introducing substantial noise to the data. Nevertheless, integration of the datasets from both institutions using batch correction approaches (Extended Data Fig. 5b-c) confirmed the low abundance of *PI16*⁺ RCs in tonsils of pediatric patients (Extended data Fig. 5d). These data further strengthen the conclusion that the tonsillar *PI16*⁺ RC compartment develops during adulthood and is further activated and remodeled during inflammatory processes.

The analysis of the extended datasets of tonsillitis patients confirmed the induction and amplification of distinct inflammation-induced processes in *PI16*⁺ RCs (Fig. 6a-b). In addition, the 'Response to FGF' pathway was significantly upregulated in the amended dataset. Cellular interaction patterns have been re-analyzed using CellChat based on the extended datasets including 81,997 T and B cells from 2 adult OSA patients and 3 adult tonsillitis patients and 23,704 FRCs from 5 adult OSA and 4 adult tonsillitis patients.

The pathway analysis of the extended datasets confirmed the major interaction patterns of PI16⁺ RCs during homeostasis and inflammation (Fig. 6 c-j and Extended Data Fig. 8).

In sum, the analysis of additional scRNA-seq data from tonsillitis patients support the conclusion that PI16⁺ RCs productively interact with T and B cells through distinct signaling pathways (Fig. 6l).

Specific points

- 3) *Throughout the manuscript it is unclear how reproducible the observed differences between adult and pediatric tonsils are. What is the variation between donors?*

We have amended the analysis of relative cell type abundance including additional datasets from tonsillitis patients (Kantonsspital St. Gallen) and data from UPenn. Statistical analysis of relative cluster abundance revealed significant differences for FRCs but not for any other stromal cell type (Extended Data Fig. 5a). The analysis of the combined KSSG and UPenn datasets (3 additional patients) confirmed the low abundance of PI16⁺ RCs in pediatric patients and further elevated PI16⁺ RCs abundance in tonsillitis patients (Extended Data Fig. 5d). Overall, these data confirm that the subepithelial PI16⁺ RC niche in human tonsils is present in adults and further expanded during inflammation.

- 4) *The manuscript relies heavily on bioinformatics analyses. Unfortunately, on several occasions, the text describing the results suggests differences that for non-bioinformatics experts are difficult to reconcile with the figures. For instance: “The transcriptional profile within FRC populations changed profoundly in adult tonsils (Fig. 4b and c) compared to the pediatric tonsils (Fig. 4a).” (lines 227-229). These profound changes are not easily detected in the UMAP representations and it would help the reader if the authors could explain in more detail the differences they are referring to.*

We have amended the text and have highlighted the respective adult FRC subpopulation in adult OSA and tonsillitis patients with arrows:

- 5) *Due to the current selection of images, conclusions on interactions between FBLN1 stromal cells and T and B cells appear to be based on 2 images re-used in multiple figures. Figure 5C is re-used extended fig 5b, the same goes for figures 5e and extended figure 5c. The panels in figure 5f are enlargements of the panels in extended figure 5d. The use of different micrographs in different figures would help to provide a sense of reproducibility of the co-localization.*

We thank the reviewer for suggesting improvements for the display of the confocal microscopy data. Fig. 5e shows the subepithelial FBLN1⁺ FRC niche that has been reconstructed in 3D to visualize surface contact areas between T/B cells and FBLN1⁺ FRCs. The original Extended Data Fig. 5c had shown the high-resolution image. We have now amended Extended Data Fig. 5c-d with images from two additional patients to provide further illustration of the co-localization of FBLN1⁺ FRCs with T and B cells (new Extended Data

Figure 6h and j). Panels in Fig. 5f represent magnifications of Figure 5e (as declared in the figure legend) and visualize cell surface contact areas (blue) that have been calculated based on fluorescence overlay between FBLN1 and CD3 or CD20 channels, respectively, which is indicative for intercellular membrane contact. Enlarged images of two additional patients have been included in the amended Extended Data Fig 6. Together, these data provide evidence for the physical interaction of PI16⁺ RCs with T and B cells in the subepithelial niche.

Reviewer 2:

In this study De Martin et al., perform a detailed characterization of the stromal populations in human palatine tonsils. Although human tonsils have been used in research for many years, the stromal cell populations and their organization has not been comprehensively studied. This manuscript includes well performed immunofluorescence microscopy revealing some new marker features of the squamous epithelium overlying the tonsil, including showing a basal layer of crypt epithelium expresses PDPN. A four marker FACS panel is developed to permit quantitation of the major stromal cell types. Comparison of pediatric and adult tonsil reveals a reduction in the PDPN+ACTA2+ subset with aging. scRNAseq of CD45-CD235a- tonsil cells is used to interrogate the gene expression profile of the different stromal types. Interestingly, the ACTA2+ population that contains perivascular (smooth muscle) cells also contains cells with FRC properties (termed perivascular reticular cells or PRC). Assessment of cells from one adult tonsillitis patient (compared to cells pooled from 5 adult OSA patients and 3 pediatric OSA patients) revealed an increase in FRC and PRC content. PI16-expressing RCs were found to make up a fraction of the FRCs (confirmed by IF) and to be enriched in adult tonsil and undergo changes in gene expression during tonsillitis. GSEA indicated these cells may have functions related to angiogenesis, cell stress response and BMP signaling. Finally, scRNAseq is performed on lymphocytes from two adult OSA patients and the tonsillitis patient. All the data were then used in CellChat analysis to examine the likely types of communications of stromal cells with each other and with lymphocytes. These data provided support for some communications that have been genetically and functionally defined in the mouse, and they suggest some unstudied communications such as maintenance of PI16+ RC by lymphocyte TGF β -derived signals. Overall, this work adds a significant amount of new descriptive data and some new insight beyond the multitude of past studies on tonsil anatomy and lymphoid composition. The evidence that adventitial PI16+ RC may not be precursor cells is notable and likely of general interest as it contrasts with the suggestion made in Buechler et al., 2021 (ref 37). A general concern I have with the study is that no attempt is made to use in vitro (or mouse model) experiments to confirm any of the suggested functional relationships identified by the scRNAseq approaches.

Specific concerns

- 1. The title and abstract imply some previously undefined subepithelial compartment has been identified. However, the compartments described appear in line with those detailed in prior studies. What is newly defined is greater heterogeneity in the FRCs than indicated by prior work, in particular PI16+ RCs and PRCs, within these lymphoid regions.*

We thank the reviewer for the insightful comments and highly productive suggestions for the improvement of the analysis and display of the data. Indeed, our study reveals the hitherto underappreciated heterogeneity of the human tonsillar stromal cell landscape and provides an in-depth analysis of age- and inflammation-associated phenotypical and functional adaptations of various cell types and subsets. We appreciate that Reviewer 1 acknowledges our focus on two FRC subsets (PRCs and PI16 RCs), which had been left understudied.

2. Can we be confident that data obtained from a single tonsillitis patient are representative of common changes during inflammation?

We have been able to acquire additional samples from three patients suffering from tonsillitis: two patients undergoing partial tonsillectomy due to bacterial infection and one patient who underwent emergency tonsillectomy due to EBV-associated tonsillitis. The patient characteristics of all patients included in the scRNA-seq analyses are listed in the new Extended Data Table 1. The scRNA-seq analysis with the three additional datasets comprises now 86,966 CD45⁻ CD235a⁻ stromal cells from three pediatric OSA patients, five adult OSA and four adult tonsillitis patients. The data and display items in Figures 3, 4 and 6 have been amended accordingly.

The analysis of the extended dataset revealed a high consistency with the previous analysis with respect to stromal cell types, FRC subset definition, transcriptional programs during inflammation and interaction patterns of PI16⁺ RCs with T and B cells. Please also see the extended reply to comment #2 by Reviewer 1 who had raised the same concern.

3. The scRNAseq data, as with past data studies by this group, has been deposited as primary data as permitted by the ArrayExpress database. However, the more widely used GEO database requires processed data to be deposited. It can be challenging for labs that lack advanced computing power to download the raw data and run the 10x cellranger pipeline themselves, especially if they are just wanting to quickly query the data. It is strongly recommended that processed data are deposited. Moreover, many studies that involve largely scRNAseq data acquisition and analysis include a user-friendly portal to facilitate data querying. Such a portal is likely to increase the utility and citation of the study.

We thank the reviewer for the kind suggestion to improve the accessibility of our scRNA-seq data. Indeed, the analysis of unprocessed primary data can be challenging and requires advanced computing capabilities. In order to facilitate easy exploration of the processed data, we have developed a web-based platform (<https://immbiosg.github.io/FRCdataExplorer/>) for interactive data exploration through an R shiny server. The processed data from this project as well as data from other projects of our group can now be queried via this platform. In addition, we have uploaded the processed data files of this project as R objects to figshare where they can be downloaded and used for further downstream analysis or exploration in any in-house pipeline after publication of the study. All links to access and explore the data and code are included in the 'Data availability' statement at line 713 of the manuscript.

4. Gene expression data are used extensively to make statements indicating a functional activity has been established. For example, on line 257 'PI16+ RCs exhibited an array of functions' and on lines 469-471 'we found PI6+ RCs utilize canonical immune cell-stimulating pathways...to increase immune cell activation'. While it may be reasonable to use mRNA expression data to suggest that cells may have certain functions, they are not sufficient to conclude that they have those functions.

Since PI16⁺ RCs are a novel FRC subset, we have used gene set enrichment and interaction analyses based on single cell transcriptomics data to assess phenotype and functional properties of the cells. Following the productive comments of this and the other reviewers, we have extensively characterized the functional properties of PI16⁺ RCs and their response to a number of inferred stimulatory mediators. These validation experiments confirmed the functional properties of PI16⁺ RCs and their ability to respond to TGFβ1, TGFβ3, FGF2 and TNFSF14 (LIGHT). Please also see our detailed response to comment #1 of Reviewer 1.

5. No attempt is made in the results or discussion to put the findings in the context of the Mourcin et al., study (ref 29) that used markers such as CD49a to facilitate tonsil FRC subsetting. This lack of sufficient credit for prior work is even more notable when it is considered that Mourcin et al. included some cell culture experiments to test functional properties of tonsil FRC subsets such as their ability to respond to TGFβ.

Our single cell transcriptomics data from three pediatric OSA patients, five adults OSA and four adult tonsillitis patients indicate that the gene encoding for CD49a (*ITGA1*) is expressed mainly in ACTA2⁺ PRCs, while only a small proportion of VSMCs and TRCs express this gene. As depicted in Figure 2 for the attention of the reviewer, CD49a (*ITGA1*) expression does not appear to be a suitable marker to clearly distinguish FRC subsets in human tonsillar tissues. It is possible that CD49a could serve as a marker for FRC subsets in human lymph nodes. Since our work is focused on tonsillar FRC, we prefer not to further extend the discussion of potential FRC markers in other human lymphoid organs.

We have credited the pioneering work of Mourcin et al. using in vitro culture of human lymphoid organ fibroblasts (line 388 and following.)

Fig. R2. ITGA1 expression in human palatine tonsils. a, d, UMAP representation of CD45⁻ CD235a⁻ stromal cells (a) and re-embedded FRCs (d). b, e, Feature UMAPs visualize ITGA1 expression pattern in different stromal cell types (b) and FRC subsets (e). c, f, Violin plots depicting ITGA1 expression level in distinct stromal cell types (c) and FRC subsets (f).

6. This study reports a lack of marginal reticular cells (MRC) based on scRNAseq. A lack of MRC in tonsil was previously reported by Mourcin et al., based on a lack of TRANCE staining (the conventional approach to identify MRC; TRANCE staining in human LN was confirmed.)

Indeed, using immunofluorescence analyses Mourcin et al. (ref. 29) have shown that TRANCE/RANKL signals are absent in areas of human tonsils that most likely harbor MRCs. Our study reveals the absence of MRCs based on comprehensive single cell transcriptomics datasets obtained from pediatric and adult palatine tonsils. We have amended the discussion and credit the work of Mourcin et al. accordingly (line 458-459).

7. The findings should also be related to the work of Heesters et al., JEM 2021 characterizing FDC from human tonsil. Based on a brief Pubmed search, the lack of literature citation extends to other studies. For example, FGFs are suggested to contribute to FRC maintenance in an autocrine manner. Yet a study showing that tonsil stromal cell proliferation is promoted by FGF5 is not cited (Park et al., 2016 PMID 27224250). PMID 27907202 also contains relevant information on FACS analysis of tonsil stromal cells and their isolation and response to inflammatory stimuli.

We thank the reviewer for the productive suggestions to amend the discussion and to incorporate additional references. We have added the references of Heesters 2021 (ref. 30) and Bar-Ephraim 2016 (ref. 41) to the reference list. We have tried to focus the Discussion section on the novel aspects of our study, i.e. (i) identification of PRCs and PI16+ RCs, (ii) structural organization of tonsillar microenvironments, and (iii) interaction of PI16+ RCs with T and B cells. Certainly, the growing literature on human FRCs deserves a comprehensive review and interpretation. We feel that the Discussion section of this study is not the

best suitable medium for an overview and in-depth appraisal of the literature dealing with human FRC biology.

8. It is commented “Our single cell transcriptomics analysis did not provide evidence that specialized, microfold (M) cell-like epithelial cells, as suggested in previous studies 17, 18, are part of the lymphoreticular epithelium.” This sentence is poorly worded. The data in ref 17 shows convincing evidence of M cells in human tonsil epithelium. This discrepancy needs more discussion. PMID 20141577 may also be relevant.

We thank the reviewer for the productive suggestions to amend the discussion around M cells. Our data are consistent with the notion that the reticulated epithelium in the crypts generates a multi-layered membranous structure that serves as a sieve for the liquids present in the oropharynx. Active transport of antigenic material through M cells, which are a part of the single cell epithelial layer in the gut, is thus not the mode of accession of immunologically active substances to the underlying tonsillar lymphoid tissues. The recommended references have been incorporated in the text in lines 461 and following.

Minor

1. UEA1 is introduced as an endothelial marker but the initial images show the epithelial expression. The text could be changed for clarity.

We thank the reviewer for thoroughly assessing consistency of our writing. We have amended the text accordingly.

2. Line 102, inlet should be inset.

The text has been changed accordingly.

3. Figure 5e and Extended data Figure 5c are the same image. Also, Extended data Figure 5d seems redundant with what is shown in Figure 5e and f.

Fig. 5e shows the subepithelial FBLN1⁺ FRC niche that has been reconstructed in 3D to visualize surface contact areas between T/B cells and FBLN1⁺ FRCs. Extended data Fig. 5c shows the original high-resolution image. We have amended Extended Data Fig. 5c-d with histological images from two additional patients to provide further evidence for the physical interaction of Pi16⁺/FBLN1⁺ FRCs with T and B cells (Extended Data Figure 6 f-g). Please see also our response to comment #5 of Reviewer 1 who had raised a similar concern.

Reviewer 3:

Ludewig and colleagues use imaging, flow cytometry and scRNAseq to characterise the human tonsil in a high level of detail, with a specific focus on the non-hematopoietic cells that are essential for orchestrating immune responses in secondary lymphoid tissues. The strength of this paper is the focus on stromal cells in human secondary lymphoid organs, which are not given the appropriate amount of attention, and the depth of the phenotyping of these cells. The weaknesses of this paper are that it is descriptive and most of the interesting insights from scRNAseq are not followed up.

We thank the reviewer for appreciating the strength of our study and for providing thoughtful comments.

Major points to consider:

The images presented in figure 1 are very beautiful and provide a nice description of the tonsil. The challenge with this type of data (where one only representative image is shown) is to present data from multiple individuals to convey how typical these features are. One approach is to quantify key features from the imaging in multiple people and show these data, or to include additional images from multiple people in the extended figures.

To address this point and to confirm that the images represent the typical topology of palatine tonsils, we have included histology from additional patients for the displayed features in the amended Extended Data Fig. 1 and 2. In particular, we have added confocal microscopy analyses of one additional patient's tonsil to Extended Data Fig. 1 d and e highlighting the topology of surface and crypt epithelium. Images of two additional patients in Extended Data Fig. 2 a,d underpin our analysis of the topology of lymphoid tissue stromal cells in the B cell zone/germinal center (Extended data Fig. 2 b-c and e-f) and the T cell zone (new Extended data Fig. 2 g and h).

Likewise, we have amended Extended Data Fig. 6 a,b and d with histological analyses of additional patients for each condition (i.e., adult OSA, adult tonsillitis and pediatric OSA). Extended Data Fig. 6 f,g and h-k have been supplemented with confocal microscopy images from two additional adult OSA patients showing that PI16 can be detected mainly intracellularly. Moreover, these data provide further visual evidence for PI16 and FBLN1 co-localization and the physical interaction of FBLN1⁺ FRCs with T and B cells.

The use of the word ageing is problematic. Its most common use pertains to older age or "elderly". Here, there is a nice comparison of paediatric and adult tonsil samples, but none of the adults can be over 53 years of age. Therefore, the age-based comparison is more relevant to childhood vs. adulthood, rather than the ageing process per se. I would recommend that the use of the word ageing is reconsidered so that the key messages are clear.

We very much appreciate the comment concerning the choice of the term "ageing". We have now used the phrase "age-dependent" to indicate that the presence of PI16⁺ RCs is related to the age of the individual.

The statistical analyses in Figure 2c-e and Extended data 2 are not appropriate. Each symbol is a single tonsil, with two coming from each person. The Mann-Whitney assumes that all datapoints are independent, which they are not as two points come from the same person. It would be more appropriate to have one point per person (possibly the average of the two tonsils). Also, for this data type the standard deviation is more appropriate to show that the SEM, as the latter is relevant for showing the spread of different populations, rather than different individuals.

We thank the reviewer for the careful review of statistical tests used for our data analysis. Indeed, in the flow cytometric analysis, data points are not fully independent, as two measurements have been performed from the same individual (left and right tonsil). Therefore, data from left and right tonsils of each

patient have been averaged and the Mann-Whitney test has been used for statistical analysis. Moreover, SD instead of SEM is now shown in the revised figure. Noteworthy, the fractions of EpCs, LECs and PDPN⁺ ACTA2⁻ FRCs did not differ between pediatric and adult OSA patients, whereas PDPN⁺ ACTA2⁺ perivascular myofibroblasts were significantly more abundant in pediatric tonsils (Fig. 2e). BECs do no longer show significantly different relative abundance between adult and pediatric OSA patients. The text of the manuscript has been revised accordingly.

Figures 3, 4 and 6 utilize single cell RNAseq to further characterize the stromal cell populations in the tonsils. Further defining changes between children and adults, and using one sample, someone with tonsillitis. With the exception of staining PI16⁺ cells in the tonsil by confocal there is no validation or further exploration of these data. Potential interacting partners are identified bioinformatically, which is interesting, but with no biological follow up this is more hypothesis generating than providing new knowledge about how tonsillar cells interact with each other. Some in vitro co-culture studies, or following up the TGFβ observations with some phospho-staining (e.g. figure 3 of this paper: <https://www.ncbi.nlm.nih.gov/pmc/articles/PMC4183221/>) would really raise the interest of these observations.

Please note that Reviewer 1 and Reviewer 2 have raised the same concern. Here, our response to comment #1 of Reviewer 1:

To establish whether the predicted ligands/stimulatory factors affect tonsillar FRCs, we first expanded bulk tonsillar fibroblasts and stimulated the cells in vitro with recombinant human TGFβ1, TGFβ3, LIGHT (TNFSF14) and FGF2. We found that TGFβ1 strongly affects PI16⁺ RC differentiation as revealed by qRT-PCR assessment of *PI16* expression in tonsillar fibroblasts from eight OSA patients (new Fig. 6k). Likewise, TGFβ1 induced the expression of *FN1* (encoding for fibronectin), a factor that was predicted to be highly expressed in PI16⁺ RCs and to be upregulated during inflammation (new Extended Data Fig. 9a). TGFβ1 and LIGHT fostered IL-6 secretion by tonsillar fibroblasts, whereas FGF2 down-regulated IL-6 production (new Extended Data Fig. 9b). In addition, LIGHT induced upregulation of ICAM surface expression, whereas FGF2 reduced ICAM expression (new Extended Data Fig.9c). Stimulation with both TGFβ1 and LIGHT showed that these factors balance the inflammatory reaction of tonsillar fibroblasts as exemplified by down-regulation of ICAM surface expression in comparison to LIGHT stimulation alone (new Extended Data Fig. 9c). Noteworthy, FGF2 appears to antagonize the effects of TGFβ1 on *PI16/FN1* expression and of LIGHT on IL-6 production and ICAM expression (new Fig. 6k and new Extended Data Fig. 9a-c). Overall, these data confirm that TGFβ, FGF and LIGHT are important factors that determine the phenotype and function of tonsillar fibroblasts.

To further assess to what extent TGFβ1, TGFβ3, LIGHT and FGF2 specifically shape the subepithelial niche generated by PI16⁺ RCs, we have sorted PI16⁺ RCs and TRCs as a control population from adult OSA and tonsillitis patients (N=5). The approach using CD34 and PDPN staining in combination with CD31, UEA1 and CD45 labeling of endothelial, epithelial and hematopoietic cells, respectively, is depicted in the new Extended Data Fig. 9d. Sorted cells were expanded in culture for 10-20 days and their cell identity and morphology was assessed using *PI16* qRT-PCR (new Extended Data Fig. 9e) and confocal microscopy (Fig. R1 for the attention of the reviewer). The in vitro cultured FRC subsets were stimulated with TGFβ1, TGFβ3, LIGHT (TNFSF14) and FGF2 proteins for 48 h. We found that FGF2 serves as a proliferation factor for TRCs and PI16⁺ RCs (new Extended Data Fig. 9f). LIGHT stimulation yielded activation of both TRCs and PI16⁺ RCs leading to increased ICAM surface expression and IL-6 production (new Extended Data Fig. 9g and h). In contrast, TGFβ3-dependent stimulatory effects appeared to be restricted to PI16⁺ RCs leading to increased IL-6 production (new Extended Data Fig. 9h). Moreover, both TGFβ1 and TGFβ3 induced *THBS1* expression mainly in PI16⁺ RCs as predicted by the bioinformatics analysis (new Extended Data Fig. 9i).

In sum, the in vitro analyses using bulk preparations of tonsillar fibroblasts and sorted PI16⁺ RCs and TRCs underscore the conclusion that TGFβ signaling serves as one of the key pathways that control the establishment of the PI16⁺ RC niche. Moreover, the data confirm the predicted activities of FGF and TNFSF14 on expansion and activation of PI16⁺ RCs, respectively.

Minor points to consider:

Please update Extended table 1 to have each patient as a row, and include the age (in years), sex and reason for tonsillectomy for each person.

Extended Data Table 1 has been replaced with a table including age, sex and reason for tonsillectomy for all patients that have been included for scRNA-seq analysis.

Extended figure 2. There appears to be a compensation issue with CD45-APC-Cy7 and ACTA2 APC, can this be resolved in flowjo to improve data quality. This will be important for any automated grouping analyses like the UMAP analysis in figure 2. The gate on the live cells is also very generous (fixable viability 510) is there a reason for this? Could the data be improved with a stricter gating here.

We very much appreciate this expert comment. We have further refined the compensation of CD45-APCCy7 and ACTA2 eFluor660 and have introduced more stringent gating as suggested by the Reviewer (Extended Data Figure 3a). The relative abundance of stromal cell types has been re-calculated based on the adjusted gating with only minor and insignificant changes as depicted in Fig. 2c-e. Whereas the fractions of EpCs, LECs, BECs and PDPN⁺ ACTA2⁻ FRCs were not affected by the age differences in OSA patients, PDPN⁺ ACTA2⁺ perivascular myofibroblasts were still significantly more abundant in pediatric tonsils (Fig. 2e).

Decision Letter, first revision:

12th Dec 2022

Dear Burkhard,

We have now finished reviewing your revised manuscript entitled "PI16+ reticular cells in human palatine tonsils activate T and B cells in distinct subepithelial niches", reference number NI-A34043A. As with the companion manuscript, NI-A34044A, we were able to obtain 2 of the 3 original referees, who while finding the study had been improved with revision, referee #1 could not fully endorse publication of the study in Nature Immunology. Referee #1 states "My main critiques of the manuscript have not been addressed. Through the way it is written (including the title and conclusions), the manuscript suggests the identification of a stromal cell subset that activates T and B cells in tonsils. This is not the case. There are no experiments in the study addressing this point." Hence the referee suggests that the data are currently over-interpreted for concluding that still over-interpreting their data by implying that the PI16+ stromal cells are "functionally" for activating the B and T cells within the human tonsil niche.

[REDACTED]

I hope that you continue to consider Nature Immunology for your results most significant for the immunology community and wish you well in your future investigations.

Kind regards,

Laurie

Laurie A. Dempsey, Ph.D.
Senior Editor
Nature Immunology
l.dempsey@us.nature.com
ORCID: 0000-0002-3304-796X

Reviewers' comments:

Reviewer #1 (Remarks to the Author):

In their revised manuscript, De Martin, Stanossek and co-workers provide novel experiments and samples to support the major conclusions of their study that the PI16+ fibroblasts form a specialized niche for T and B cell activation in activated tonsils.

While the initial manuscript was based on a single case of tonsillitis, the analysis of additional inflamed tonsils enables conclusions to be drawn on this part of the manuscript.

Unfortunately, the study remains purely descriptive and does not contain any experimental validation of the main conclusion that PI16+ stromal cells are activators of T and B cells. The authors have

added in vitro stimulation experiments that show the ability of ligand-receptor pairs identified by the in silico interactome prediction to affect FRC transcriptomes. These types of experiments provide evidence that the tested ligands could have a function in shaping the differentiation and phenotype of FRC subsets. They do not provide evidence of a functional role for PI16+ cells as activators of immune cells.

In my opinion, this manuscript provides a beautiful overview of the stromal cell landscape in human tonsils at the in situ and molecular levels. In addition, it now provides clues on the signals that might shape and differentiate such environments.

These are valuable additions to our knowledge and understanding of the structural backbone of the immune system that are unnecessarily diluted by unsubstantiated statements on the functionality of the stromal cells and their role as activators of T and B cells.

As in my initial review, I still feel that most of the conclusions, as well as the title of the study, are worded too strongly and are not supported by the data presented.

While the title states that "PI16+ reticular cells in human palatine tonsils activate T and B cells in distinct subepithelial niches", the manuscript does not contain any evidence to prove this. The fact that there is transcription of a cytokine (il-6) or expression of an adhesion molecule (ICAM) is not proof for T and B cell activation. In lieu of changing the title and text, authors would first need to provide evidence that PI16+ FRC, or at least stromal cells stimulated in such a way as to resemble PI16+ cells, are capable of activating T or B cells. Second, evidence is needed to show that T and B cells with the activated phenotype from the co-culture experiments co-localize with PI16+ cells in situ.

The discussion contains various speculative statements presented as conclusions that need to be reworded to reflect better that they are not supported by experiments. This includes all statements on interactions with lymphocytes and modes of activation of the FRC. What is discussed are possible interactions, possible functionalities, and possible importance. Transcriptome does not equal function.

The final conclusion: "our study shows that PI16+ RCs underpin and sustain mucosal immune responses in the human oropharynx through the formation of an adaptive subepithelial niche for activated T and B cells in palatine tonsils" is a good example as neither the underpinning nor the sustaining of a niche is proven. This needs to be removed.

Minor points:

From the methods section, it is unclear what the "change in RNA" means in the qPCR experiments in figure 6K and Suppl fig 9. Was this corrected for the increased number of cells in the conditions with FGF2?

Line 322: "Noteworthy, FGF2 appears to antagonize the effects of TGFβ1 on PI16/FN1 expression." I could not find this data (stimulation with FGF2 + TGFB1) in the figures. Can the authors please check for the correctness of this statement?

Reviewer #3 (Remarks to the Author):

The authors have addressed all my concerns, I appreciate their thorough and thoughtful response.

Appeal Decision Letter

Dear Burkhard,

Thank you for supplying your rebuttal to the referees comments and to our editorial concerns to your manuscript entitled "PI16+ reticular cells in human palatine tonsils activate T and B cells in distinct subepithelial niches". Please revise your manuscript as we discussed previously in our email correspondence.

We therefore invite you to revise your manuscript taking into account all reviewer and editor comments. Please highlight all changes in the manuscript text file in Microsoft Word format.

Once you have made these revisions, please use the URL below to submit the revised manuscript with figures, an updated life science reporting summary and any supplemental checklists, and a point-by-point response addressing the reviewers' criticisms.

The Reporting Summary can be found here:

The Editorial Policy Checklist can be found here: <https://www.nature.com/documents/nr-editorial-policy-checklist.pdf>

[REDACTED]

Please let us know how you wish to proceed and when we can expect your revised manuscript.

With kind regards,

Laurie

Laurie A. Dempsey, Ph.D.
Senior Editor
Nature Immunology
l.dempsey@us.nature.com
ORCID: 0000-0002-3304-796X

Author Rebuttal, first revision

See Inserted PDF

Dear Laurie, Dear Editors of Nature Immunology,

We would like to thank the Editors for the opportunity to revise our manuscript now entitled “PI16⁺ reticular cells in human palatine tonsils govern T cell activity in distinct subepithelial niches”. We are pleased that Reviewer 3 found that the comprehensive work of the first revision had satisfied all of their comments. We are also glad that Reviewer 1 has appreciated the major part of the first revision highlighting the work as “*beautiful overview of the stromal cell landscape in human tonsils*” and “*valuable additions to our knowledge and understanding of the structural backbone of the immune system*”. We acknowledge the remaining concerns of Reviewer 1, which are addressed in a point-by-point manner below. In brief, we have experimentally addressed the major concern of Reviewer 1 by co-culturing PI16⁺ RCs with autologous tonsillar CD3⁺ T cells. Based on these novel results and following the recommendation of the reviewer, we have adapted conclusions concerning functions of PI16⁺ RCs.

- 1) In their revised manuscript, De Martin, Stanossek and co-workers provide novel experiments and samples to support the major conclusions of their study that the PI16⁺ fibroblasts form a specialized niche for T and B cell activation in activated tonsils.
- 2) While the initial manuscript was based on a single case of tonsillitis, the analysis of additional inflamed tonsils enables conclusions to be drawn on this part of the manuscript.

We highly appreciate that Reviewer 1 values our work that substantiates the initial description of inflammation-induced remodelling of the human tonsillar stromal cell landscape. The additional scRNA-seq data from inflamed tonsillar tissue have fully corroborated all initial conclusions.

- 3) Unfortunately, the study remains purely descriptive and does not contain any experimental validation of the main conclusion that PI16⁺ stromal cells are activators of T and B cells. The authors have added in vitro stimulation experiments that show the ability of ligand-receptor pairs identified by the in silico interactome prediction to affect FRC transcriptomes. These types of experiments provide evidence that the tested ligands could have a function in shaping the differentiation and phenotype of FRC subsets. They do not provide evidence of a functional role for PI16⁺ cells as activators of immune cells.

We are glad that Reviewer 1 acknowledges the validation experiments performed during the first revision, which revealed factors that have refined the tonsillar FRC phenotype in terms of PI16 expression, and delineated the effects of various autocrine and paracrine factors on the phenotype and activity of bulk tonsillar FRCs, sorted TCRs and sorted PI16⁺ RCs. We found, for example, that IL-6 production by bulk FRCs, TCRs and PI16⁺ RCs was modulated differently by the tested soluble factors. Hence, the validation experiments provided in the first revision clearly showed biological effects beyond “FRC transcriptomes”.

Nevertheless, we agree with Reviewer 1 that production and modulation of established lymphocyte-activating factors, such as IL-6 or ICAM1, by human FRCs do not formally prove a functional role of PI16⁺ RCs during the interaction with particular immune cell populations. We have therefore established a co-culture system of tonsillar PI16⁺ RCs with autologous tonsillar T cells to provide further evidence for productive PI16⁺ RC-immune cell interaction.

Please note that we had sufficient autologous material, immune cells and PI16⁺ RCs, available from four (4) patients to perform the requested experiments. Further details are provided in our response to comment #6.

- 4) In my opinion, this manuscript provides a beautiful overview of the stromal cell landscape in human tonsils at the in situ and molecular levels. In addition, it now provides clues on the signals that might shape and differentiate such environments.
These are valuable additions to our knowledge and understanding of the structural backbone of the immune system that are unnecessarily diluted by unsubstantiated statements on the functionality of the stromal cells and their role as activators of T and B cells.
- 5) As in my initial review, I still feel that most of the conclusions, as well as the title of the study, are worded too strongly and are not supported by the data presented.

We are pleased with the appreciation of Reviewer 1 for our molecular and cellular analyses of human tonsillar tissues and the elaboration of tonsillar FRC-derived niche factors. We also acknowledge the request for further validation of stromal-immune cell interaction, in addition to the validation experiments provided in the first revision. For a detailed description of the co-culture experiments please see our response to comment #6.

Based on the results from the previous and novel validation experiments, we have re-phrased relevant parts of the manuscript:

- Title: *“PII6+ reticular cells in human palatine tonsils govern T cell activity in distinct subepithelial niches”*
 - Abstract (second last sentence): *“Interactome analysis combined with ex vivo and in vitro validation revealed that T cell activity within subepithelial niches is controlled by distinct molecular pathways during PII6+ RC-lymphocyte interaction.”*
 - Introduction (second last sentence, line 79): *“Subepithelial PII6+ RCs formed a distinct niche adjacent to the lymphoreticular epithelium in adult tonsils supporting direct contact with T and B cells and the provision of costimulatory and growth factors that regulate T cell activity.”*
 - Discussion (line 535): *“Moreover, our findings from the co-culture system of tonsillar PII6+ RCs with autologous T cells further support the notion that PII6+ RCs form dedicated niches that foster T cell activation and differentiation.”*
 - Discussion (line 545): *“Our data suggest that LIGHT and TGF- β 3 could serve as hitherto unknown secondary signals that drive FRC subset specification and activation under inflammatory conditions²¹. PII6+ RC differentiation appears to be fueled by the interaction with lymphocytes leading to the enhanced production of distinct niche factors such as IL-6, which further promote reciprocal stimulation of the cells.”*
 - Discussion, concluding sentence (line 552): *“In sum, our study shows that PII6+ RCs underpin an adaptive subepithelial niche in mucosal lymphoid tissues of the human oropharynx and shape the microenvironment for efficient T cell activation and differentiation.”*
- 6) While the title states that *“PII6+ reticular cells in human palatine tonsils activate T and B cells in distinct subepithelial niches”*, the manuscript does not contain any evidence to prove this. The fact that there is transcription of a cytokine (il-6) or expression of an adhesion molecule (ICAM) is not proof for T and B cell activation. In lieu of changing the title and text, authors would first need to provide evidence that PII6+ FRC, or at least stromal cells stimulated in such a way as to resemble PII6+ cells, are capable of activating T or B cells. Second, evidence is needed to show that T and B cells with the activated phenotype from the co-culture experiments co-localize with PII6+ cells in situ.

To address this concern, we have employed an in vitro co-culture system of PII6⁺ RCs and sorted tonsillar CD3⁺ T cells. In brief, CD3⁺ T cells from four OSA or tonsillitis patients were co-cultured with autologous PII6⁺ RCs for 3 days in the presence or absence of the niche factors FGF2 or TGF- β 3. To mimick conditions of T cell receptor ligation, we used phytohemagglutinin (PHA). Proliferation and activation of CD4⁺ and CD8⁺ T cells was assessed using dilution of the intracellular fluorescent dye carboxyfluorescein succinimidyl

ester (CSFE) and expression of the activation marker CD25, respectively. Moreover, we have monitored the production of the cytokines IL-6 as measure for PI16⁺ RC activity, and IL-10 as further indicator for T cell activity/differentiation. We found significant proliferation of tonsillar CD4⁺ T cells (new Fig. 6j and k) or CD8⁺ T cells (new Extended Data Fig. 10a and b) only in the presence of PI16⁺ RCs when compared to PHA stimulation alone. PHA-mediated activation of T cells in the presence of PI16⁺ RCs significantly increased expression of the T cell activation marker CD25 by CD4⁺ T cells (new Fig. 6j and l) and CD8⁺ T cells (new Extended Data Fig. 10a and c). T cell activation as determined by CD25 expression was further enhanced in the presence of TGF- β 3 (new Fig. 6l and Extended Data Fig. 10c and d). Moreover, TGF- β 3 significantly down-tuned IL-10 production by T cells (new Fig. 6m). These data further highlight that PI16⁺ RCs employ discrete signaling pathways to support T cell activation including TGF- β 3-mediated increase of T cell proliferation, CD25 upregulation and down-tuning of the anti-inflammatory cytokine IL-10. At the same time, PI16⁺ RCs maintained production of IL-6 in the presence of FGF2 or TGF- β 3 in the co-culture system (new Extended Data Fig. 10e) supporting the notion that IL-6 provision by PI16⁺ RCs may foster the activity of IL-6-responsive cells such as T and B cells, in the subepithelial niche.

- 7) The discussion contains various speculative statements presented as conclusions that need to be reworded to reflect better that they are not supported by experiments. This includes all statements on interactions with lymphocytes and modes of activation of the FRC. What is discussed are possible interactions, possible functionalities, and possible importance. Transcriptome does not equal function.

We appreciate Reviewer 1's well-founded interest in the appropriate interpretation of scientific results. We would like to highlight that we are fully aware of the fact that our interpretations based on transcriptomics data are indicative or suggestive. Hence, we have put emphasis on the correct wording in the description and interpretation of the data in the results section; i.e. always using appropriate wording such as "infer and visualize intercellular communications" (line 352), "interaction analysis predicted" (line 361) or "indicated substantially increased information flow" (line 366).

As to be anticipated and largely accepted for the discussion part of a scientific publication, we have used inductive inference to draw conclusions about potential functional relationships between tonsillar FRCs and lymphocytes. Scientific discussion, as in the revision process of this manuscript, yields improved interpretation based on productive suggestions by peers and editors. We thus gladly followed the suggestions of the reviewers and have generated additional data in two rounds of revisions and have adapted the conclusions and interpretation of data accordingly.

- 8) The final conclusion: "our study shows that PI16⁺ RCs underpin and sustain mucosal immune responses in the human oropharynx through the formation of an adaptive subepithelial niche for activated T and B cells in palatine tonsils" is a good example as neither the underpinning nor the sustaining of a niche is proven. This needs to be removed.

We have re-phrased the concluding sentence as follows: "In sum, our study shows that PI16⁺ RCs underpin an adaptive subepithelial niche in mucosal lymphoid tissues of the human oropharynx and shape the microenvironment for efficient T cell activation and differentiation."

- 9) Minor points:

From the methods section, it is unclear what the "change in RNA" means in the qPCR experiments in figure 6K and Suppl fig 9. Was this corrected for the increased number of cells in the conditions with FGF2?

We thank the reviewer for the careful reading and apologize for the incorrect labelling of the Y-axis in the respective items. Relative mRNA expression was calculated by the comparative cycling threshold method ($\Delta\Delta$ -CT method) using the expression of Glyceraldehyde-3-phosphate dehydrogenase (GAPDH) for

normalization. Fold change of relative mRNA expression compared to medium control was calculated. The methods section has been amended accordingly and the figure items have been corrected.

Line 322: “Noteworthy, FGF2 appears to antagonize the effects of TGFβ1 on PI16/FN1 expression.” I could not find this data (stimulation with FGF2 + TGFβ1) in the figures. Can the authors please check for the correctness of this statement?

We thank the reviewer for pointing out the confusing labelling. The corrected sentence now reads as follows: “Noteworthy, FGF2 appears to antagonize the effects of LIGHT on IL-6 production and ICAM1 expression (Extended Data Fig. 9b-c).”

Decision Letter, second revision:

9th Mar 2023

Dear Burkhard,

Thank you for submitting your revised manuscript "PI16+ reticular cells in human palatine tonsils govern T cell activity in distinct subepithelial niches" (NI-A34043C). I have looked over your revised manuscript and your point-by-point response to the remaining comments from referee #1. These revisions are satisfactory, thus we'll be happy in principle to publish it in Nature Immunology, pending minor revisions to comply with our editorial and formatting guidelines.

We will now perform detailed checks on your paper and will send you a checklist detailing our editorial and formatting requirements in about a week. Please do not upload the final materials and make any revisions until you receive this additional information from us.

If you had not uploaded a Word file for the current version of the manuscript, we will need one before beginning the editing process; please email that to immunology@us.nature.com at your earliest convenience.

Thank you again for your interest in Nature Immunology. Please do not hesitate to contact me if you have any questions. {Please note, however, that I'll be traveling away from the office for the next 10 days - as I'll be in Taiwan for conference & lab visits - hence I might not be able to respond right away.}

Kind regards,

Laurie

Laurie A. Dempsey, Ph.D.
Senior Editor
Nature Immunology
l.dempsey@us.nature.com
ORCID: 0000-0002-3304-796X

Final Decision Letter:

Dear Burkhard,

I am delighted to accept your manuscript entitled "PI16+ reticular cells in human palatine tonsils govern T cell activity in distinct subepithelial niches" for publication in an upcoming issue of Nature Immunology.

Over the next few weeks, your paper will be copyedited to ensure that it conforms to Nature Immunology style. Once your paper is typeset, you will receive an email with a link to choose the appropriate publishing options for your paper and our Author Services team will be in touch regarding any additional information that may be required.

Please note that *Nature Immunology* is a Transformative Journal (TJ). Authors may publish their research with us through the traditional subscription access route or make their paper immediately open access through payment of an article-processing charge (APC). Authors will not be required to make a final decision about access to their article until it has been accepted. [Find out more about Transformative Journals](https://www.springernature.com/gp/open-research/transformative-journals).

Your paper will be published online soon after we receive your corrections and will appear in print in the next available issue. Content is published online weekly on Mondays and Thursdays, and the embargo is set at 16:00 London time (GMT)/11:00 am US Eastern time (EST) on the day of publication. Now is the time to inform your Public Relations or Press Office about your paper, as they might be interested in promoting its publication. This will allow them time to prepare an accurate and satisfactory press release. Include your manuscript tracking number (NI-A34043D) and the name of the journal, which they will need when they contact our office.

About one week before your paper is published online, we shall be distributing a press release to news organizations worldwide, which may very well include details of your work. We are happy for your institution or funding agency to prepare its own press release, but it must mention the embargo date

and Nature Immunology. Our Press Office will contact you closer to the time of publication, but if you or your Press Office have any enquiries in the meantime, please contact press@nature.com.

Also, if you have any spectacular or outstanding figures or graphics associated with your manuscript - though not necessarily included with your submission - we'd be delighted to consider them as candidates for our cover. Simply send an electronic version (accompanied by a hard copy) to us with a possible cover caption enclosed.

If you have not already done so, we strongly recommend that you upload the step-by-step protocols used in this manuscript to the Protocol Exchange. Protocol Exchange is an open online resource that allows researchers to share their detailed experimental know-how. All uploaded protocols are made freely available, assigned DOIs for ease of citation and fully searchable through nature.com. Protocols can be linked to any publications in which they are used and will be linked to from your article. You can also establish a dedicated page to collect all your lab Protocols. By uploading your Protocols to Protocol Exchange, you are enabling researchers to more readily reproduce or adapt the methodology you use, as well as increasing the visibility of your protocols and papers. Upload your Protocols at www.nature.com/protocolexchange/. Further information can be found at www.nature.com/protocolexchange/about .

Please note that we encourage the authors to self-archive their manuscript (the accepted version before copy editing) in their institutional repository, and in their funders' archives, six months after publication. Nature Portfolio recognizes the efforts of funding bodies to increase access of the research they fund, and strongly encourages authors to participate in such efforts. For information about our editorial policy, including license agreement and author copyright, please visit www.nature.com/ni/about/ed_policies/index.html

Kind regards,

Laurie

Laurie A. Dempsey, Ph.D.
Senior Editor
Nature Immunology
l.dempsey@us.nature.com
ORCID: 0000-0002-3304-796X